# Incomplete Multi-View Clustering via Neighborhood-Conditioned Diffusion

**Qian Guo** [1][2] **Gaohui Zuo** [1] **Bingbing Jiang** [3] **Guangrui Fan** [1] **Zhihua Cui** [1][2] **Xinyan Liang** [4] **Jianjian Ding** [1]

## Abstract

Incomplete multi-view clustering (IMVC) aims to uncover shared clustering structures from heterogeneous views with partial observations. Recently, existing generative IMVC methods have made significant progress in this field; however, they still remain limited in two aspects. On the one hand, they rely on weak cross-view signals, resulting in unstable latent recovery when facing missing data. On the other hand, they overlook stable cross-view neighborhood structures, leading to weak structural constraint. To address these limitations, we propose neighborhood-conditioned diffusion for incomplete multi-view clustering (IMVC-NCD), which achieves robust latent completion. Our method learns compact view-specific latent representations and constructs a unified conditioning vector by aggregating stable local neighborhood structures from available views while encoding missingness states, providing reliable guidance for diffusion-based denoising. With neighborhood-level conditioning, IMVC-NCD produces semantically aligned and view-consistent latent representations that are well suited for clustering, even under high missing-view ratios. Extensive experiments on four benchmark datasets demonstrate the effectiveness and robustness of our method compared with state-of-the-art IMVC approaches. Our code is available at https://github.com/zgh1115/IMVC-NCD.

---

[1]School of Computer Science and Technology, Taiyuan University of Science and Technology, Taiyuan, Shanxi Province, China. [2]Shanxi Key Laboratory of Big Data Analysis and Parallel Computing, Taiyuan University of Science and Technology, Taiyuan, Shanxi Province, China. [3]School of Information Science and Engineering, Hangzhou Normal University, Hangzhou, China. [4]Institute of Big Data Science and Industry, Key Laboratory of Evolutionary Science Intelligence of Shanxi Province, Shanxi University, Taiyuan, China. Correspondence to: Xinyan Liang <liangxinyan48@163.com>.

*Proceedings of the $43^{rd}$ International Conference on Machine Learning*, Seoul, South Korea. PMLR 306, 2026. Copyright 2026 by the author(s).

## 1. Introduction

Advances in sensing and data acquisition technologies have enabled many applications to be represented by heterogeneous multi-view features (Liang et al., 2022; 2025a;b; Yuan et al., 2024b; Guo et al., 2024), making multi-view learning a central topic in unsupervised representation learning (Chowdhury et al., 2025; Jiang et al., 2025b; Liu et al., 2024). Multi-view clustering (MVC) exploits cross-view consistency to uncover shared structures (Moujahid & Dornaika, 2025), yet most methods assume fully observed views (Liu et al., 2025; Zhou et al., 2025; Yang et al., 2025; Zhang et al., 2026), an assumption often violated in practice due to missing-views. This motivates the development of robust incomplete multi-view clustering (IMVC) methods.

Existing IMVC methods generally fall into two paradigms: imputation-free methods (Jiang et al., 2025a; Dai et al., 2025; Wu et al., 2025; Xu et al., 2024) and imputation-based methods (Yuan et al., 2025; Jin et al., 2025; Dong et al., 2025). The imputation-free methods avoid explicit recovery by directly aggregating available views, making them simple and efficient in handling incomplete observations. However, such aggregation often suffers from severe information imbalance and fails to capture shared semantics under high missing ratios, since critical cross-view information may be absent. To alleviate these issues, imputation-based methods reconstruct missing-views to explicitly complete the multi-view information, which can better preserve cross-view consistency and improve representation quality for downstream clustering.

As a category of imputation-based methods, generative models have emerged as a promising direction for addressing IMVC, owing to their ability to model complex data distributions and recover latent representations of missing-views. Early research employed generative adversarial networks (GANs)-based, which introduce adversarial learning to synthesize missing-view features, typically by jointly minimizing reconstruction loss and discriminator loss to enhance recovery quality (Wang et al., 2023; Shu et al., 2024; Peng et al., 2025). However, GAN-based methods often face challenges such as training instability and potential model collapse. Subsequently, flow-based generative methods came into researchers' view (Hirschorn & Avidan, 2023; Tan et al., 2024; Xu et al., 2025). For instance, BURG (Jin et al.,

2025) recovers missing-views by modeling cross-view relationships. However, its reliance on a rigid Gaussian prior assumption limits its flexibility in capturing complex, non-Gaussian intrinsic data structures. Recently, researchers have begun to explore diffusion-based generative methods, which can iteratively optimize noisy representations and exhibit strong robustness against severe data incompleteness (Daras et al., 2023). For example, Wen et al. (Wen et al., 2024) utilized conditional diffusion to generate missing-views, while Zhang et al. (Zhang et al., 2025) combined diffusion with contrastive learning to enhance latent representations oriented towards clustering. Nevertheless, existing diffusion-based methods ignore stable cross-view neighborhood structures, leading to weak structural constraint and unstable completion under missing-view scenarios.

We observe that under missing-view scenarios, local neighborhood relationships remain relatively stable, providing reliable structural cues for latent space recovery. Based on this insight, we propose neighborhood-conditioned diffusion for incomplete multi-view clustering (IMVC-NCD). First, IMVC-NCD employs view-specific autoencoders to compress high-dimensional raw data into low-dimensional latent representations, preserving view-specific characteristics while reducing computational complexity. Second, a neighborhood-conditioned construction module is introduced to encode missing-view patterns into a unified conditioning vector, thereby aggregating cross-view neighborhood information. Next, guided by this condition, a conditional diffusion model performs progressive latent denoising to recover missing-view representations in a stable and view-consistent manner. Finally, through synergistic coupling of neighborhood-aware conditioning and diffusion-based generation, IMVC-NCD alleviates the constraint insufficiency problem in missing-view recovery. This method generates discriminative latent features with enhanced cluster separability, directly improving performance in downstream clustering tasks.

The overall contributions of this work are summarized as follows:

- We propose a neighborhood-conditioned diffusion framework that unifies view-specific latent learning and structure-aware generative recovery for IMVC, enabling robust modeling under missing-view patterns.
- We introduce neighborhood-conditioned guidance that mitigates the limitations of weak paired correspondences and global latent cues, enabling stable and structure-aware diffusion-based latent completion.
- Extensive experiments demonstrate that IMVC-NCD consistently outperforms state-of-the-art methods across varying missing ratios, with notably improved robustness under severe incomplete-view settings.

## 2. Related Work

### 2.1. Deep Incomplete Multi-View Clustering

Deep learning has played a significant driving role in the development of IMVC. Currently, deep IMVC (DIMVC) methods can generally be divided into two categories: imputation-free and imputation-based frameworks. Imputation-free methods directly aggregate representations from available views without explicitly reconstructing missing ones, and perform clustering on the aggregated representations (Dai et al., 2025; Wu et al., 2025; Xu et al., 2024). In contrast, imputation-based methods explicitly recover missing-views before clustering (Yuan et al., 2025; Jin et al., 2025; Dong et al., 2025). According to their imputation mechanisms, existing DIMVC approaches can be further categorized into three groups: (i) graph-structured methods (Dong et al., 2025; Chen et al., 2025a; Liu et al., 2023), which leverage graph structures or cross-view relational information to propagate knowledge from complete views to incomplete ones; (ii) prototype-based methods (Yuan et al., 2025; Jin et al., 2025; Yuan et al., 2024a), which learn prototypes from observed views and exploit sample–prototype relationships to infer missing-view; (iii) generative model-based methods, including GAN-based methods (Wang et al., 2023; Shu et al., 2024; Peng et al., 2025), VAE-based methods (Gao & Pu, 2025; Cai et al., 2023), flow-based methods (Jin et al., 2025), and diffusion-based methods (Wen et al., 2024; Zhang et al., 2025), which employ deep generative models to capture the joint distribution of multi-view data and synthesize or reconstruct missing-views for subsequent clustering.

### 2.2. Diffusion Model

Diffusion models (Ho et al., 2020) have emerged as a powerful class of generative models that learn complex data distributions through a forward–reverse noising process. By gradually corrupting data with Gaussian noise and then learning to invert this process, diffusion models can synthesize or recover high-fidelity representations with remarkable stability (Nichol & Dhariwal, 2021; Ma et al., 2025). Compared with other types of generative models, diffusion models avoid mode collapse and training instability while offering strong likelihood modeling capabilities (Dhariwal & Nichol, 2021). These properties make diffusion models well-suited for incomplete multi-view learning (Wen et al., 2024; Zhang et al., 2025), because recovering missing-views requires strong and reliable generative priors to guide the reconstruction. However, while diffusion models provide a flexible generative mechanism, their performance heavily depends on the quality of conditional signals used to guide the denoising trajectory (Ho & Salimans, 2021). Such conditions may take various forms, including labels, features, or auxiliary structural cues. Therefore, designing effective and structurally reliable conditions becomes essential when

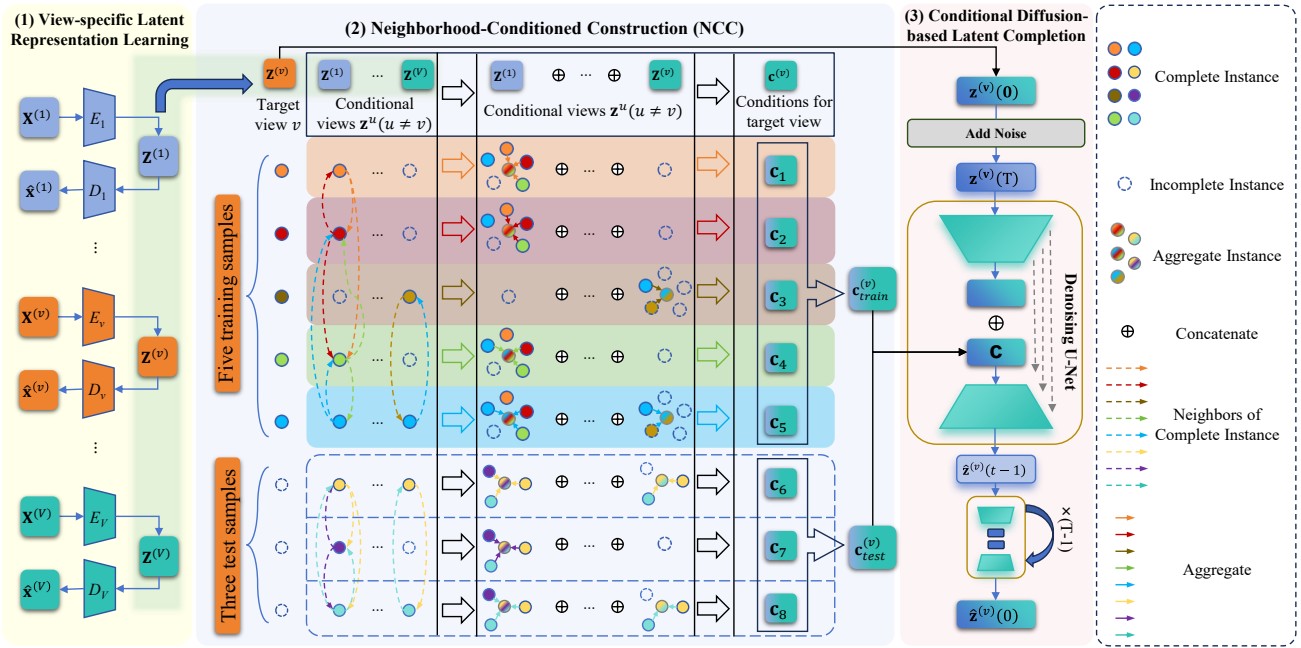

Figure 1. The framework of IMVC-NCD, illustrated using a representative example with $V$ views and 8 samples. The IMVC-NCD consists of three main modules: (1) **view-specific latent representation learning**; (2) **neighborhood-conditioned construction (NCC)**; and (3) **conditional diffusion-based latent completion**. Note that $E$ and $D$ denote the encoder and decoder, respectively, and $\mathbf{c}^v_{train}$ and $\mathbf{c}^v_{test}$ represent the conditions for the target view $v$ utilized during training and inference, respectively.

applying diffusion models to IMVC.

## 3. Method

This section details the architecture of our proposed IMVC-NCD framework. As illustrated in Fig. 1, the model is composed of three modules: view-specific latent representation learning, neighborhood-conditioned construction (NCC), and conditional diffusion-based latent completion.

**Notations.** We denote the incomplete multi-view data as $\{\mathbf{X}^{(v)}\}_{v=1}^V$, where $V$ is the number of views. Each view $\mathbf{X}^{(v)} = \{\mathbf{x}_1^{(v)}, \mathbf{x}_2^{(v)}, \cdots, \mathbf{x}_N^{(v)}\} \in \mathbb{R}^{N \times d_v}$ consists of $N$ instances with dimensionality $d_v$. To explicitly represent the missing-pattern distribution across views, we define a binary mask set $\mathbf{M} = \{\mathbf{m}^{(v)}\}_{v=1}^V$, where $\mathbf{m}^{(v)} = \{m_1^{(v)}, m_2^{(v)}, \cdots, m_N^{(v)}\} \in \mathbb{R}^N$. Each mask entry $m_i^{(v)}$ indicates whether the $i$-th sample is observed in the $v$-th view:

$$m_i^{(v)} = \begin{cases} 1, & \text{if } i\text{-th sample is observed in } v\text{-th view,} \\ 0, & \text{otherwise.} \end{cases}$$
(1)

### 3.1. View-Specific Latent Representation Learning

Deep autoencoders have been widely adopted in IMVC for unsupervised representation learning (Chen et al., 2025b; Yan et al., 2025; Jin et al., 2025; Yuan et al., 2025; Dong et al., 2025). Through nonlinear transformations and

reconstruction-driven learning, they effectively model heterogeneous, high-dimensional multi-view data and produce stable latent representations that serve as a reliable foundation for subsequent condition construction and diffusion-based latent completion.

In our framework, the autoencoder is implemented in a view-specific manner to accommodate the heterogeneity across different feature spaces. For each view $v$, the encoder $E_v(\cdot)$ maps the input sample $\mathbf{x}_i^{(v)}$ into a shared latent space of dimension $d_z$, producing the latent representation

$$\mathbf{z}_i^{(v)} = E_v(\mathbf{x}_i^{(v)}), \quad \mathbf{Z}^{(v)} \in \mathbb{R}^{N \times d_z}.$$
(2)

The encoder is designed as a multi-layer perceptron (MLP) with nonlinear activations, enabling it to capture complex structures in each view while aligning them into a unified latent domain. Correspondingly, the decoder $D_v(\cdot)$ reconstructs input features from the latent representations via

$$\hat{\mathbf{x}}_i^{(v)} = D_v(\mathbf{z}_i^{(v)}).$$
(3)

The autoencoder parameters are jointly optimized by minimizing the reconstruction loss across all views:

$$\mathcal{L}_{\text{rec}} = \sum_{v=1}^V \sum_{i=1}^N \left\| \hat{\mathbf{x}}_i^{(v)} - \mathbf{x}_i^{(v)} \right\|_2^2.$$
(4)

Ultimately, the model learns compact and structurally consistent latent representations, which serve as a critical pre-

requisite for the subsequent neighborhood-conditioned construction and diffusion-based completion stages.

## 3.2. Neighborhood-Conditioned Construction

In the process of applying diffusion models to IMVC, a paramount challenge lies in devising conditional signals that are rich in information and reliable, so as to better guide denoising trajectory. Unlike image or text generation tasks with well-defined semantic conditions, incomplete multi-view data exhibit heterogeneous feature spaces and irregular missing patterns, making it essential to construct conditions that capture cross-view structural cues while remaining robust to missingness. Without such conditions, diffusion-based latent completion becomes weakly constrained, leading to unreliable and view-inconsistent recovery.

We found that local neighborhood structures encode stable cross-view semantics and are relatively robust to noise and view-specific distortions. Motivated by this observation, we propose the NCC module, which transforms available latent representations into a unified conditioning vector by aggregating neighborhood information from each available view and representing missing-views with zero-valued conditional components, thus forming a robust multi-view condition. Specifically, for a target view $v$, we first obtain latent embeddings $\mathbf{Z}^{(v)}$ and extract the latent sets $\mathbf{Z}^{(u)}$ for all conditional views $u \neq v$, together with their missingness indicators. For each sample $i$, NCC constructs the per-view conditional component $\mathbf{c}_i^{(u)}$ according to three possible availability states.

**(1) Fully available condition:** If view $u$ is observed for sample $i$, NCC performs a $k$-nearest-neighbor (kNN) search among all valid samples of that view, including sample itself, and computes a neighborhood-aggregated prototype as

$$\mathbf{c}_i^{(u)} = \frac{1}{k} \sum_{j \in \mathcal{N}_k^{(u)}(i)} \mathbf{z}_j^{(u)}. \tag{5}$$

Here, $\mathcal{N}_k^{(u)}(i)$ denotes the index set of the $k$ nearest neighbors of sample $i$ in the latent space of view $u$, including $i$ itself, identified among all valid samples of that view.

**(2) Fully missing condition:** If all conditional views of sample $i$ are missing, that is, $m_i^{(u)} = 0$ for every conditional view $u \neq v$, the NCC can not compute any neighborhood-aggregated prototype because there is no valid latent representation on the conditional side. In this case, NCC assigns zero vectors to all conditional components, resulting in a zero-valued conditioning input that corresponds to unconditional diffusion. Although such samples offer no cross-view information, they remain essential during training. Prior work on classifier-free diffusion guidance (CFG) (Ho & Salimans, 2021) shows that incorporating unconditional samples helps diffusion models avoid overfitting to specific

conditioning patterns, reduces training bias, and encourages the model to learn the marginal data distribution. Inspired by this insight, exposing the diffusion model to fully missing conditional views enables it to capture the intrinsic latent structure of the target view $v$ without relying on auxiliary guidance. This unconditional learning pathway improves robustness under diverse missingness patterns and acts as an implicit regularizer for the diffusion model.

**(3) Partially available condition:** If a sample $i$ has partially observed conditional views, NCC computes $\mathbf{c}_i^{(u)}$ for each available view via kNN aggregation, while representing missing-views with zero-valued components to preserve a fixed conditioning structure. Specifically, the conditional component per-view is assigned as

$$\mathbf{c}_i^{(u)} = \begin{cases} \dfrac{1}{k} \displaystyle\sum_{j \in \mathcal{N}_k^{(u)}(i)} \mathbf{z}_j^{(u)}, & \text{if } m_i^{(u)} = 1, \\ \mathbf{0}, & \text{if } m_i^{(u)} = 0. \end{cases} \tag{6}$$

This mixed pattern exposes the diffusion model to partially informative conditioning signals. Samples with partially available conditional views are especially useful: the observed views provide realistic cross-view cues for missing-view prediction, while the zero-filled components serve as a light regularizer that prevents over-reliance on fully informative conditions. As a result, partially available conditions naturally sit between conditional and unconditional training, helping the model learn to denoise under incomplete guidance and improving robustness and generalization in practical incomplete-view scenarios.

Finally, for the $i$-th sample of the target view $v$, we construct the unified conditioning vector $\mathbf{c}_i^{(v)}$ by concatenating the per-view conditional components in a fixed order:

$$\mathbf{c}_i^{(v)} = \text{Concat}\big(\{\mathbf{c}_i^{(u)} \mid u \neq v\}\big). \tag{7}$$

By aggregating the unified conditioning vectors across all samples, we define the condition for the target view $v$ as:

$$\mathbf{c}^{(v)} = \big[\mathbf{c}_1^{(v)}, \mathbf{c}_2^{(v)}, \ldots, \mathbf{c}_N^{(v)}\big]^\top. \tag{8}$$

During training, we adopt a classifier-free conditional dropout strategy by replacing $\mathbf{c}^{(v)}$ with a zero vector with probability $p$, enabling unconditional denoising. This prevents the model from becoming overly reliant on the neighborhood conditions, compelling it to learn the intrinsic data distribution and ensuring robust denoising even when structural cues are ambiguous.

Overall, NCC provides a principled way to convert, incomplete multi-view observations into stable, semantically aligned conditioning signals suitable for diffusion-based latent completion.

**Algorithm 1** Training of IMVC-NCD

---

**Input:** Incomplete multi-view data $\{X^{(v)}\}_{v=1}^{V}$, mask $\{m^{(v)}\}$, autoencoder (AE) epochs $e_{\text{AE}}$, diffusion epochs $e_{\text{diff}}$, diffusion steps $T$, schedule $\{\alpha_t\}_{t=1}^{T}$.
**Output:** $\hat{y}$.

1: **for** $e = 1$ to $e_{\text{AE}}$ **do**
2:     Update encoders $E_v$ and decoders $D_v$ with Eq. (4).
3: **end for**
4: Obtain latent $\{Z^{(v)}\}_{v=1}^{V}$ by Eq. (2).
5: **for** $v = 1$ to $V$ **do**
6:     Construct NCC condition $\mathbf{c}^{(v)}$ using Eqs. (5)–(8).
7:     **for** $e = 1$ to $e_{\text{diff}}$ **do**
8:         Sample $t \sim \{1, \ldots, T\}$ and $\epsilon \sim \mathcal{N}(0, I)$.
9:         Form $\mathbf{z}^{(v)}(t)$ with Eq. (9).
10:        Train DDPM using Eq. (13).
11:     **end for**
12:     Initialize $\mathbf{z}^{(v)}(T) \sim \mathcal{N}(\mathbf{0}, \mathbf{I})$.
13:     **for** $t = T$ down to 1 **do**
14:         Sample reverse noise $\boldsymbol{\xi} \sim \mathcal{N}(\mathbf{0}, \mathbf{I})$ and perform the reverse denoising step using Eqs. (14)–(16).
15:     **end for**
16:     Obtain the recovered latent representations of missing samples in view $v$ as $\hat{\mathbf{z}}^{(v)}(0)$.
17: **end for**
18: Obtain the recovered latent representations for all views, concatenate them for multi-view fusion, and run $k$-means to obtain $\hat{y}$.

---

### 3.3. Conditional Diffusion-Based Latent Completion

After constructing the neighborhood-conditioned condition for the target view, denoted as $\mathbf{c}^{(v)}$, our goal is to recover the missing latent representation of the target view using a conditional denoising diffusion probabilistic model. Originally, denoising diffusion probabilistic models (DDPM) was introduced as a generative framework that learns to recover structured data, such as images, from progressively corrupted noise through a step-by-step denoising process. In our method, rather than generating images in the original data space, we transfer this mechanism to the latent space for IMVC, where the goal is to recover the missing latent representation of a target view from noise under the guidance of neighborhood-conditioned information. Specifically, DDPM defines a forward process that gradually adds Gaussian noise and a learned reverse process that iteratively denoises the latent variable to recover a clean representation (Ho et al., 2020). This module follows the standard DDPM formulation and extends it by injecting the NCC-produced condition into each denoising step.

**Forward (Noise Addition) Process.** During the forward diffusion process, we gradually add Gaussian noise to the clean latent $\mathbf{z}^{(v)}(0)$, yielding $\mathbf{z}^{(v)}(t)$ that becomes increas-

ingly noisy and approaches pure Gaussian noise:

$$
\begin{aligned}
q(\mathbf{z}^{(v)}(t) \mid \mathbf{z}^{(v)}(0)) = \\
\mathcal{N}\Big(\mathbf{z}^{(v)}(t); \sqrt{\bar{\alpha}_t}\, \mathbf{z}^{(v)}(0),\, (1 - \bar{\alpha}_t)\mathbf{I}\Big),
\end{aligned}
\tag{9}
$$

where $\mathcal{N}(\cdot)$ denotes the Gaussian distribution, $\alpha_t$ and $\bar{\alpha}_t$ are defined as $\alpha_t = 1 - \beta_t$ and $\bar{\alpha}_t = \prod_{s=1}^{t} \alpha_s$, with $t \in \{1, \ldots, T\}$, where $T$ is the total number of diffusion steps, and $\beta_{1:T} \in (0, 1)$ controls the noise variance.

**Reverse (Denoising) Process.** The reverse denoising process is a Markov Chain that runs backward from $\mathbf{z}^{(v)}(T)$ to $\mathbf{z}^{(v)}(0)$. Specifically, each reverse step is modeled as a Gaussian transition:

$$
\begin{aligned}
p_\theta(\mathbf{z}^{(v)}(t-1) \mid \mathbf{z}^{(v)}(t), \mathbf{c}^{(v)}) = \\
\mathcal{N}\Big(\mathbf{z}^{(v)}(t-1); \boldsymbol{\mu}_\theta(\mathbf{z}^{(v)}(t), t, \mathbf{c}^{(v)}), \sigma_t^2\mathbf{I}\Big),
\end{aligned}
\tag{10}
$$

where $\boldsymbol{\mu}_\theta(\cdot)$ and $\sigma_t$ denote the mean and standard deviation of the Gaussian transition, respectively, both of which are defined in Eq. (11) and Eq. (12).

$$
\begin{aligned}
\boldsymbol{\mu}_\theta(\mathbf{z}^{(v)}(t), t, \mathbf{c}^{(v)}) = \\
\frac{1}{\sqrt{\alpha_t}}\left(\mathbf{z}^{(v)}(t) - \frac{1 - \alpha_t}{\sqrt{1 - \bar{\alpha}_t}}\, \varepsilon_\theta(\mathbf{z}^{(v)}(t), t, \mathbf{c}^{(v)})\right),
\end{aligned}
\tag{11}
$$

$$
\sigma_t = \sqrt{\frac{1 - \bar{\alpha}_{t-1}}{1 - \bar{\alpha}_t}\,(1 - \alpha_t)}.
\tag{12}
$$

Here, $\varepsilon_\theta(\cdot)$ implements using a U-Net network that predicts the noise $\varepsilon$ added at step $t$, conditioned on the target-view condition $\mathbf{c}^{(v)}$ constructed by the NCC module. $\theta$ represents the model parameters. By incorporating $\mathbf{c}^{(v)}$ into the denoising process, the diffusion model is guided to recover latent representations that are view-consistent and semantically aligned with the available views.

**Training Objective.** The model is trained by minimizing the standard noise-prediction loss:

$$
\mathcal{L}_{\text{diff}} = \mathbb{E}_{t, \mathbf{z}^{(v)}(0), \varepsilon}\left[\left\|\varepsilon - \varepsilon_\theta(\mathbf{z}^{(v)}(t), t, \mathbf{c}^{(v)})\right\|_2^2\right],
\tag{13}
$$

where $\varepsilon \sim \mathcal{N}(0, \mathbf{I})$. During training, $\mathbf{c}^{(v)}$ is directly provided by NCC, which naturally handles fully-available, partially-available, and fully-missing conditions.

**Missing-View Generation.** During the inference stage, we recover a missing latent representation by initializing $\mathbf{z}^{(v)}(T) \sim \mathcal{N}(0, \mathbf{I})$ and performing iterative denoising from $t = T$ down to 1. To better leverage the cross-view NCC condition $\mathbf{c}^{(v)}$, while maintaining stability when conditional cues are weak or noisy, we adopt CFG during sampling.

*Table 1.* Clustering results on the HandWritten dataset under different missing rates (30%, 50%, 70%). The best results are highlighted in **bold**, and the second-best results are underlined.

| Method | ACC (%) | | | NMI (%) | | | Purity (%) | | |
|---|---|---|---|---|---|---|---|---|---|
| | 30% | 50% | 70% | 30% | 50% | 70% | 30% | 50% | 70% |
| BSV | $51.49 \pm 2.29$ | $38.24 \pm 2.25$ | $27.15 \pm 1.31$ | $47.01 \pm 1.71$ | $32.21 \pm 1.00$ | $19.48 \pm 0.69$ | $53.69 \pm 1.54$ | $39.54 \pm 2.04$ | $27.76 \pm 1.09$ |
| Concat | $55.48 \pm 1.57$ | $42.19 \pm 0.99$ | $28.31 \pm 2.75$ | $51.66 \pm 0.99$ | $38.24 \pm 1.59$ | $23.50 \pm 0.95$ | $57.32 \pm 1.15$ | $44.21 \pm 0.98$ | $30.45 \pm 0.80$ |
| SIMVC-SA | $82.44 \pm 2.52$ | $78.24 \pm 2.45$ | $61.15 \pm 4.79$ | $71.60 \pm 2.10$ | $63.99 \pm 1.73$ | $48.88 \pm 3.23$ | $82.44 \pm 2.52$ | $78.29 \pm 2.41$ | $61.94 \pm 4.24$ |
| RecFormer | $93.07 \pm 0.41$ | $91.74 \pm 0.43$ | $84.43 \pm 1.18$ | $86.12 \pm 0.64$ | $83.39 \pm 0.91$ | $72.23 \pm 1.08$ | $93.07 \pm 0.41$ | $91.74 \pm 0.43$ | $84.43 \pm 1.18$ |
| CPSPAN | $89.70 \pm 1.02$ | $89.16 \pm 1.21$ | $83.75 \pm 2.93$ | $82.38 \pm 1.03$ | $81.35 \pm 1.51$ | $\underline{77.09 \pm 0.71}$ | $89.70 \pm 1.02$ | $89.16 \pm 1.21$ | $83.75 \pm 2.93$ |
| RPCIC | $89.90 \pm 0.12$ | $89.77 \pm 1.03$ | $73.87 \pm 1.61$ | $82.39 \pm 0.57$ | $82.15 \pm 1.15$ | $72.26 \pm 1.81$ | $89.90 \pm 0.12$ | $89.77 \pm 1.03$ | $77.77 \pm 1.30$ |
| DMVG | $\underline{93.55 \pm 0.02}$ | $91.66 \pm 0.29$ | $\underline{86.77 \pm 0.60}$ | $\underline{86.89 \pm 0.51}$ | $\underline{83.77 \pm 0.40}$ | $75.71 \pm 0.86$ | $\underline{93.55 \pm 0.02}$ | $91.66 \pm 0.29$ | $\underline{86.77 \pm 0.60}$ |
| BURG | $86.60 \pm 0.71$ | $76.90 \pm 1.02$ | $79.05 \pm 0.89$ | $77.87 \pm 0.64$ | $75.66 \pm 0.34$ | $71.74 \pm 0.70$ | $86.60 \pm 0.71$ | $78.90 \pm 0.51$ | $79.05 \pm 0.89$ |
| PMIMC | $89.03 \pm 0.18$ | $77.81 \pm 1.86$ | $76.43 \pm 3.69$ | $81.32 \pm 0.52$ | $76.82 \pm 1.91$ | $72.60 \pm 1.47$ | $89.03 \pm 0.18$ | $80.28 \pm 1.57$ | $78.20 \pm 3.18$ |
| $A^2$CLN | $92.48 \pm 0.04$ | $\underline{91.95 \pm 0.86}$ | $82.80 \pm 2.05$ | $85.45 \pm 0.52$ | $83.56 \pm 0.91$ | $76.87 \pm 2.47$ | $92.48 \pm 0.04$ | $\underline{91.95 \pm 0.86}$ | $82.80 \pm 2.05$ |
| **IMVC-NCD (ours)** | $\mathbf{94.56 \pm 0.12}$ | $\mathbf{92.68 \pm 0.05}$ | $\mathbf{87.55 \pm 0.15}$ | $\mathbf{88.46 \pm 0.11}$ | $\mathbf{85.34 \pm 0.21}$ | $\mathbf{77.13 \pm 0.21}$ | $\mathbf{94.56 \pm 0.12}$ | $\mathbf{92.68 \pm 0.05}$ | $\mathbf{87.55 \pm 0.15}$ |

Specifically, conditional and unconditional noise predictions are combined as:

$$\tilde{\varepsilon}_\theta(\mathbf{z}^{(v)}(t),\, t,\, \mathbf{c}^{(v)}) = \\ (1+\omega)\,\varepsilon_\theta(\mathbf{z}^{(v)}(t),\, t,\, \mathbf{c}^{(v)}) - \omega\,\varepsilon_\theta(\mathbf{z}^{(v)}(t),\, t), \quad (14)$$

where $\varepsilon_\theta(\mathbf{z}^{(v)}(t), t, \mathbf{c}^{(v)})$ and $\varepsilon_\theta(\mathbf{z}^{(v)}(t), t)$ denote the conditional and unconditional noise predictions, respectively, and $\omega \geq 0$ is the guidance scale controlling the strength of conditional guidance.

We then compute the guided mean by replacing $\varepsilon_\theta$ in Eq. (11) with $\tilde{\varepsilon}_\theta$:

$$\tilde{\boldsymbol{\mu}}_\theta(\mathbf{z}^{(v)}(t),\, t,\, \mathbf{c}^{(v)}) = \\ \frac{1}{\sqrt{\alpha_t}} \left( \mathbf{z}^{(v)}(t) - \frac{1-\alpha_t}{\sqrt{1-\bar{\alpha}_t}}\,\tilde{\varepsilon}_\theta(\mathbf{z}^{(v)}(t),\, t,\, \mathbf{c}^{(v)}) \right). \quad (15)$$

Finally, the denoising update is

$$\hat{\mathbf{z}}^{(v)}(t-1) = \tilde{\boldsymbol{\mu}}_\theta(\mathbf{z}^{(v)}(t),\, t,\, \mathbf{c}^{(v)}) + \sigma_t\,\boldsymbol{\xi}, \\ \boldsymbol{\xi} \sim \mathcal{N}(0, \mathbf{I}), \quad (16)$$

where $\hat{\mathbf{z}}^{(v)}(t-1)$ denotes the recovered latent representation of the target view $v$ after the denoising update at step $t-1$, $\sigma_t$ is the standard deviation defined in Eq. (12), and $\boldsymbol{\xi}$ denotes a Gaussian noise term independently sampled at each denoising step. After $T$ denoising steps, the final output $\hat{\mathbf{z}}^{(v)}(0)$ is taken as the recovered latent representation of the target view. More implementation and experimental details are provided in the appendix.

The overall optimization workflow of IMVC-NCD is summarized in Algorithm 1. This workflow adopts a two-stage training protocol, where the autoencoders are first pretrained and then fixed to provide a stable latent space for DDPM training. This avoids drifting latent representations and neighborhood relations during diffusion learning, thereby ensuring stable conditional guidance. After completing the optimization, we concatenate the recovered latent representations of missing-views with the existing ones and apply $k$-means to obtain the final clustering results.

*Table 2.* Statistics of the datasets.

| Dataset | Samples | Views | Classes | Features |
|---|---|---|---|---|
| Handwritten | 2000 | 5 | 10 | 76/216/64/240/47 |
| Scene-15 | 4485 | 3 | 15 | 20/50/59 |
| Fashion | 10000 | 3 | 10 | 784/784/784 |
| Aloi_deep | 10800 | 3 | 100 | 2048/4096/2048 |

## 4. Experiment

To evaluate the effectiveness of IMVC-NCD, we conduct extensive experiments to address the following research questions: (Q1) Does IMVC-NCD surpass existing state-of-the-art IMVC methods in clustering performance? (Q2) How much does each component of IMVC-NCD contribute to the overall improvements? (Q3) How do the key hyper-parameters influence the performance of IMVC-NCD? (Q4) How does IMVC-NCD improve the quality of completed representations in terms of visualization separability and cluster compactness? (Q5) What is the computational cost of IMVC-NCD in terms of runtime and memory usage? (Q6) Can IMVC-NCD maintain strong performance under different missing-view scenarios?

**Datasets and Metrics.** We evaluate IMVC-NCD on four widely used multi-view benchmarks, whose statistics are summarized in Table 2: HandWritten (LeCun et al., 1989), Scene-15 (Fei-Fei & Perona, 2005), Fashion (Xiao et al., 2017), and Aloi_deep (Liu et al., 2023). For a comprehensive analysis, we adopt three widely used clustering metrics, including Normalized Mutual Information (NMI), Accuracy (ACC), and Purity. More Details see Appendix.

**Implementation Details.** We use the Adam optimizer (learning rate $1 \times 10^{-4}$) for models. The autoencoders are pretrained for 200 epochs. We adopt full-batch training for the autoencoders to stabilize optimization and ensure consistent reconstruction across views. Subsequently, each view-specific DDPM is trained for 500 epochs with a batch size of 512 and $T = 1000$ diffusion steps, employing a lin-

*Table 3.* Clustering results on three benchmark datasets (missing rates 10%, 30%, 50%). The best results are highlighted in **bold**, and the second-best results are underlined.

| | Method | ACC (%) | | | NMI (%) | | | Purity (%) | | |
|---|---|---|---|---|---|---|---|---|---|---|
| | | 10% | 30% | 50% | 10% | 30% | 50% | 10% | 30% | 50% |
| Fashion | BSV | 51.54 ± 4.25 | 41.05 ± 2.73 | 33.02 ± 1.34 | 50.44 ± 2.42 | 38.41 ± 1.18 | 27.70 ± 1.55 | 55.70 ± 3.26 | 43.76 ± 1.91 | 35.25 ± 1.11 |
| | Concat | 62.33 ± 7.19 | 54.00 ± 3.70 | 35.24 ± 2.68 | 69.78 ± 3.54 | 58.01 ± 3.71 | 41.36 ± 3.17 | 69.37 ± 5.57 | 58.18 ± 3.89 | 40.53 ± 3.28 |
| | SIMVC-SA | 72.79 ± 1.02 | 66.20 ± 0.57 | 59.58 ± 1.47 | 72.22 ± 0.86 | 65.16 ± 0.60 | 56.08 ± 1.83 | 76.09 ± 1.15 | 70.07 ± 0.78 | 61.55 ± 1.72 |
| | RecFormer | 63.01 ± 4.98 | 68.87 ± 3.88 | 68.28 ± 1.08 | 62.97 ± 3.44 | 67.53 ± 2.34 | 64.20 ± 0.87 | 64.25 ± 4.69 | 70.43 ± 2.52 | 68.48 ± 1.00 |
| | CPSPAN | 72.89 ± 3.76 | 72.33 ± 5.58 | 67.10 ± 6.11 | 77.11 ± 1.30 | 76.15 ± 2.16 | 72.20 ± 3.02 | 76.18 ± 3.20 | 76.33 ± 4.30 | 71.42 ± 4.97 |
| | RPCIC | 63.12 ± 1.72 | 55.99 ± 6.54 | 58.35 ± 6.86 | 72.71 ± 1.25 | 67.83 ± 2.40 | 67.12 ± 2.23 | 68.18 ± 1.52 | 62.96 ± 6.01 | 64.42 ± 5.61 |
| | DMVG | 83.49 ± 1.53 | 82.25 ± 1.04 | 77.62 ± 0.23 | 81.27 ± 0.47 | 78.06 ± 0.85 | 72.76 ± 0.29 | 83.52 ± 1.51 | 82.31 ± 1.95 | 77.62 ± 0.23 |
| | BURG | 73.32 ± 3.80 | 52.78 ± 0.41 | 46.40 ± 1.50 | 71.92 ± 2.40 | 54.67 ± 3.08 | 41.49 ± 1.43 | 74.59 ± 2.43 | 56.74 ± 1.82 | 47.07 ± 1.58 |
| | PMIMC | 75.99 ± 4.72 | 71.34 ± 6.30 | 66.79 ± 2.02 | 78.09 ± 0.76 | 75.57 ± 2.12 | **74.04 ± 1.50** | 79.13 ± 2.73 | 75.09 ± 4.59 | 72.03 ± 2.29 |
| | A$^2$CLN | 65.29 ± 0.76 | 63.80 ± 1.53 | 73.68 ± 1.05 | 77.91 ± 0.34 | 77.28 ± 0.20 | 71.70 ± 1.47 | 70.33 ± 0.04 | 69.10 ± 0.05 | 78.27 ± 1.05 |
| | Ours | **87.56 ± 0.29** | **85.52 ± 0.15** | **80.50 ± 0.29** | **82.66 ± 0.36** | **79.25 ± 0.07** | 72.88 ± 0.04 | **87.56 ± 0.29** | **85.52 ± 0.15** | **80.50 ± 0.29** |
| Aloi_deep | BSV | 64.14 ± 1.23 | 50.63 ± 2.20 | 37.37 ± 1.34 | 81.29 ± 0.65 | 63.15 ± 0.62 | 45.36 ± 0.59 | 69.89 ± 1.13 | 55.10 ± 1.76 | 40.18 ± 1.07 |
| | Concat | 71.07 ± 2.90 | 59.60 ± 1.26 | 41.39 ± 1.30 | 89.75 ± 1.24 | 77.47 ± 0.46 | 68.63 ± 0.97 | 76.52 ± 2.39 | 64.44 ± 0.88 | 44.99 ± 1.54 |
| | SIMVC-SA | 90.26 ± 0.27 | 90.01 ± 0.52 | 87.23 ± 0.27 | 97.62 ± 0.07 | 97.43 ± 0.09 | 96.42 ± 0.09 | 90.26 ± 0.27 | 90.01 ± 0.52 | 87.23 ± 0.27 |
| | RecFormer | **93.03 ± 0.36** | 91.53 ± 0.74 | 89.30 ± 1.15 | **98.54 ± 0.07** | 98.19 ± 0.11 | 97.72 ± 0.19 | 94.78 ± 0.33 | 93.60 ± 0.44 | 92.88 ± 0.77 |
| | CPSPAN | 74.67 ± 3.35 | 74.42 ± 3.51 | 73.51 ± 2.02 | 92.84 ± 0.56 | 92.61 ± 0.73 | 92.02 ± 0.69 | 74.67 ± 3.35 | 74.42 ± 3.51 | 73.51 ± 2.02 |
| | RPCIC | 82.83 ± 2.41 | 81.57 ± 0.72 | 85.38 ± 0.95 | 95.36 ± 0.49 | 95.23 ± 0.37 | 96.17 ± 0.30 | 86.28 ± 1.62 | 85.20 ± 0.87 | 88.31 ± 0.88 |
| | DMVG | 91.47 ± 1.40 | 90.18 ± 1.63 | 92.01 ± 0.81 | 98.03 ± 0.25 | 97.71 ± 0.24 | 98.05 ± 0.27 | 93.16 ± 0.96 | 92.28 ± 1.13 | 93.63 ± 0.75 |
| | BURG | 73.52 ± 0.39 | 67.56 ± 0.30 | 67.28 ± 0.04 | 92.36 ± 0.98 | 84.80 ± 0.69 | 89.19 ± 0.18 | 78.91 ± 0.92 | 72.01 ± 0.23 | 71.89 ± 0.24 |
| | PMIMC | 85.54 ± 1.63 | 86.47 ± 1.61 | 87.17 ± 2.03 | 96.64 ± 0.31 | 96.85 ± 0.34 | 97.05 ± 0.55 | 89.03 ± 1.20 | 89.49 ± 1.29 | 90.05 ± 1.73 |
| | A$^2$CLN | 50.86 ± 0.57 | 55.66 ± 0.56 | 54.99 ± 0.69 | 78.07 ± 0.57 | 81.45 ± 1.07 | 81.24 ± 1.37 | 53.84 ± 0.29 | 57.61 ± 0.19 | 57.05 ± 0.63 |
| | Ours | **93.70 ± 0.38** | **93.28 ± 0.16** | **93.39 ± 0.07** | 98.33 ± 0.21 | **98.30 ± 0.04** | **98.19 ± 0.05** | **94.85 ± 0.44** | **94.58 ± 0.21** | **94.41 ± 0.06** |
| Scene-15 | BSV | 30.76 ± 1.13 | 26.10 ± 1.24 | 20.79 ± 0.71 | 30.93 ± 2.26 | 24.47 ± 0.57 | 17.36 ± 0.42 | 35.46 ± 1.94 | 29.61 ± 0.82 | 22.80 ± 0.60 |
| | Concat | 32.85 ± 1.04 | 26.63 ± 0.76 | 21.44 ± 0.80 | 29.89 ± 0.54 | 24.31 ± 0.64 | 19.78 ± 0.83 | 36.00 ± 0.73 | 30.57 ± 0.47 | 24.99 ± 0.83 |
| | SIMVC-SA | 41.91 ± 1.10 | 40.06 ± 0.56 | 37.39 ± 0.76 | 30.04 ± 1.01 | 25.42 ± 1.43 | 21.79 ± 0.50 | 36.05 ± 1.12 | 30.88 ± 0.88 | 27.33 ± 0.60 |
| | RecFormer | 40.74 ± 0.70 | 36.40 ± 0.70 | 31.04 ± 0.94 | 40.73 ± 0.88 | 36.41 ± 1.09 | 30.66 ± 0.80 | 46.04 ± 0.98 | 41.01 ± 1.54 | 34.89 ± 1.07 |
| | CPSPAN | 40.66 ± 2.86 | 39.15 ± 2.34 | 36.85 ± 1.77 | 38.77 ± 2.78 | 37.57 ± 1.92 | 34.24 ± 1.91 | 45.98 ± 2.69 | 43.67 ± 2.02 | 40.62 ± 1.85 |
| | RPCIC | 39.05 ± 1.17 | 36.83 ± 1.61 | 37.71 ± 1.37 | 39.16 ± 0.91 | 40.03 ± 0.54 | 40.11 ± 0.76 | 44.23 ± 1.05 | 42.29 ± 0.99 | 42.17 ± 1.17 |
| | DMVG | 43.55 ± 0.04 | 41.26 ± 0.99 | 37.41 ± 0.27 | 43.52 ± 0.35 | 40.59 ± 0.60 | 36.44 ± 0.66 | 48.18 ± 0.11 | 45.53 ± 0.95 | 43.41 ± 1.46 |
| | BURG | 40.51 ± 0.80 | 34.92 ± 0.88 | 28.12 ± 0.91 | 37.35 ± 0.40 | 32.87 ± 1.02 | 27.59 ± 0.96 | 42.56 ± 0.43 | 36.01 ± 1.03 | 31.13 ± 0.49 |
| | PMIMC | 38.72 ± 0.98 | 41.49 ± 1.03 | 39.76 ± 0.16 | 40.30 ± 0.46 | **42.73 ± 0.54** | **41.78 ± 0.04** | 42.49 ± 0.81 | 46.81 ± 1.19 | 44.89 ± 0.48 |
| | A$^2$CLN | 43.70 ± 0.76 | 39.60 ± 0.20 | 34.72 ± 1.05 | 43.29 ± 0.34 | 36.17 ± 0.20 | 34.44 ± 1.47 | 47.38 ± 0.04 | 40.25 ± 0.05 | 35.34 ± 1.05 |
| | Ours | **45.47 ± 0.55** | **43.38 ± 0.36** | **40.56 ± 0.83** | **45.01 ± 0.34** | 42.00 ± 0.15 | 37.24 ± 0.32 | **49.32 ± 0.76** | **47.71 ± 0.16** | **45.29 ± 0.30** |

*Table 4.* Ablation study of the proposed components under two missing ratios on Handwritten and Fashion datasets.

| Variants | Components | | | Handwritten (30%) | | | Handwritten (50%) | | | Fashion (10%) | | | Fashion (30%) | | |
|---|---|---|---|---|---|---|---|---|---|---|---|---|---|---|---|
| | $\mathcal{C}$ | $\mathcal{M}$ | $\mathcal{K}$ | ACC (%) | NMI (%) | PUR (%) | ACC (%) | NMI (%) | PUR (%) | ACC (%) | NMI (%) | PUR (%) | ACC (%) | NMI (%) | PUR (%) |
| Unconditional (w/o $\mathcal{C}$) | ✗ | – | – | 79.80 | 63.21 | 79.80 | 52.10 | 32.51 | 52.55 | 81.77 | 73.63 | 81.77 | 49.65 | 43.79 | 50.87 |
| (w/o $\mathcal{K}$) | ✓ | ✓ | ✗ | 82.95 | 66.18 | 82.95 | 54.80 | 35.44 | 54.80 | 77.37 | 71.56 | 77.37 | 49.71 | 43.60 | 50.48 |
| (w/o $\mathcal{M}$) | ✓ | ✗ | ✓ | 90.60 | 80.30 | 90.60 | 54.55 | 42.46 | 54.55 | 84.38 | 78.16 | 84.38 | 57.23 | 52.17 | 58.45 |
| ($\mathcal{C} + \mathcal{M} + \mathcal{K}$) | ✓ | ✓ | ✓ | 94.56 | 88.46 | 94.56 | 92.68 | 85.34 | 92.68 | 87.56 | 82.66 | 87.56 | 85.52 | 79.25 | 85.52 |

ear noise schedule. Classifier-free guidance is applied with $p = 0.1$ for training and $\omega = 1.0$ for inference, utilizing $k = 5$ nearest neighbors for conditioning. Detailed network architectures and settings are provided in the Appendix.

To construct incomplete multi-view inputs, for a given missing rate $\eta$, we independently mask $\eta \times N$ instances in each view without relying on feature values or labels. To avoid samples losing all views, we ensure that each instance retains at least one observed view by randomly restoring one when necessary. This protocol is consistently applied across all datasets for fair comparison.

### 4.1. Performance Comparisons (Q1)

We compare our IMVC-NCD with ten state-of-the-art IMVC methods. **Imputation-free methods** include SIMVC-SA (Wen et al., 2023). **Imputation-based meth-** ods include **Non-DIMVC** methods: BSV (Zhao et al., 2016) and Concat (Zhao et al., 2016). They also include **DIMVC** methods, i.e., **graph-structured**: RecFormer (Liu et al., 2023); **prototype-based**: CPSPAN (Jin et al., 2023), RP-CIC (Yuan et al., 2024a), and PMIMC (Yuan et al., 2025); and **generative-model-based**: DMVG (Wen et al., 2024), BURG (Jin et al., 2025), and A$^2$CLN (Peng et al., 2025). Notably, since DMVG introduces the DAD module as a plug-and-play component rather than a standalone clustering model, we integrate our own autoencoder into its framework to enable the generation of complete latent embeddings for subsequent clustering. More Details see Appendix.

We compare IMVC-NCD with representative state-of-the-art IMVC methods across four benchmarks. Tables 1 and 3 report the average clustering performance under different missing rates, from which three consistent observations can be drawn. (1) IMVC-NCD achieves the best overall perfor-

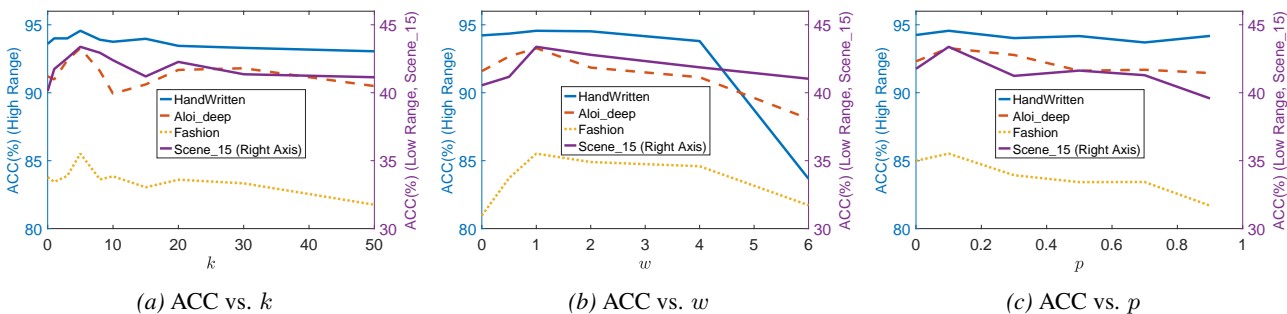

*(a)* ACC vs. $k$           *(b)* ACC vs. $w$           *(c)* ACC vs. $p$

*Figure 2*. ACC comparison under different hyperparameter settings under a missing rate of 30%: (a) varying $k$ in NCC; (b) varying guidance scale $w$; (c) varying dropout rate $p$.

mance across all datasets and missing-view ratios, consistently outperforming strong baselines, demonstrating the effectiveness of neighborhood-conditioned diffusion for latent completion. (2) Although the performance of all methods degrades as the missing rate increases, IMVC-NCD exhibits substantially stronger robustness with smaller performance drops across all datasets. (3) The substantial performance gain over DMVG validates our key observation: when views are randomly missing, local neighborhood relationships remain relatively stable, thereby providing reliable structural cues for robust latent space recovery. Overall, these results confirm that IMVC-NCD consistently outperforms existing IMVC methods while remaining robust under increasing missingness.

### 4.2. Ablation Studies (Q2)

To evaluate the contribution of each component in IMVC-NCD, we conduct ablation experiments on the Handwritten and Fashion datasets. Specifically, we decompose the model into three components: $\mathcal{C}$ (condition input component), which enables the diffusion model to take an external conditioning vector; $\mathcal{M}$ (multi-view fusion component), which aggregates latent features from all available views to form a joint condition; and $\mathcal{K}$ (kNN neighborhood component), which constructs the condition by aggregating the $k$ nearest neighbors in the latent space. Based on these components, we design three variants: "(w/o $\mathcal{C}$)", "(w/o $\mathcal{M}$)", and "(w/o $\mathcal{K}$)", which respectively correspond to removing the condition input, removing the multi-view fusion, and removing the kNN neighborhood construction. "($\mathcal{C} + \mathcal{M} + \mathcal{K}$)" keeps all components enabled.

As shown in Table 4, we can observe: (1) The performance consistently drops when any component is removed, demonstrating that $C$, $M$, and $K$ each make indispensable contributions to the overall effectiveness of IMVC-NCD. (2) Moreover, as the view missing rate increases, the performance gaps between the full model and its ablated variants become more pronounced, indicating that each component

plays an increasingly critical role in maintaining robustness under severe incompleteness. (3) Removing both the multi-view fusion $M$ and the kNN neighborhood construction $K$ leads to the most significant degradation, indicating that the proposed neighborhood-conditioned construction is crucial for providing reliable guidance to the diffusion model under missing-view scenarios.

### 4.3. Hyperparameters Analysis (Q3)

We evaluate the sensitivity of IMVC-NCD to three key hyperparameters: the neighborhood size $k$ in NCC, the classifier-free guidance scale $w$, and the dropout rate $p$. As shown in Figs. 2, the ACC remains stable across a wide range of $k$, with optimal performance typically achieved when $k$ lies between 3 and 7, indicating that IMVC-NCD is robust to the choice of neighborhood size. Similarly, the clustering performance is relatively insensitive to the guidance scale $w$ when it lies in the range of 0.7 to 1.5, and only degrades when the guidance strength becomes excessively large. In addition, the performance shows moderate sensitivity to the dropout rate $p$, remaining stable when $p$ lies in the range of 0.05 to 0.15, while overly large dropout rates lead to noticeable performance degradation. These results confirm that IMVC-NCD maintains strong stability under different hyperparameter settings.

### 4.4. Representation Quality Analysis: Visualization and Compactness (Q4)

To further verify the representation quality of IMVC-NCD, we show the t-SNE visualizations of the raw features and the completed latent representations in Fig. 3. Compared with the raw features, the learned representations exhibit clearer separation and more compact clusters, indicating improved discriminability under missing-view settings. We further measure cluster compactness by computing the average intra-cluster distance between the final concatenated representations and their assigned $k$-means centers. As shown in Fig. 4, our method achieves lower intra-cluster

distances than the baseline under a 50% missing rate, confirming that IMVC-NCD produces more cohesive cluster structures.

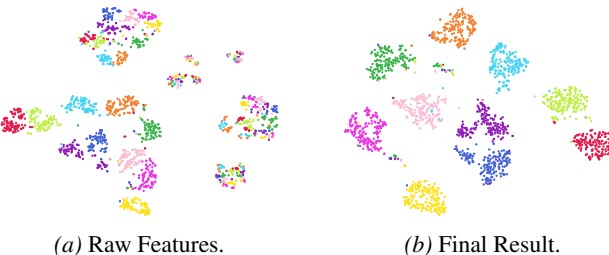

*(a)* Raw Features.     *(b)* Final Result.

*Figure 3.* t-SNE visualizations under a missing rate of 30% on HandWritten dataset.

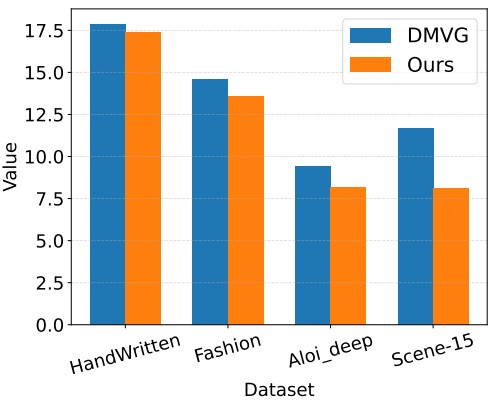

*Figure 4.* Cluster compactness analysis comparing our method with baselines under a missing rate 50%.

### 4.5. Efficiency Analysis (Q5)

To further evaluate the computational cost of IMVC-NCD, we report the runtime and memory usage of representative methods. The notation X/Y denotes runtime in seconds and GPU memory usage in GB, respectively. As shown in Table 5, diffusion-based methods generally require more time and memory. However, the current best-performing methods, including DMVG and IMVC-NCD, are both diffusion-based. Among them, IMVC-NCD has comparable runtime and memory consumption to the strong baseline DMVG, while achieving better clustering performance. This indicates that the performance improvement of IMVC-NCD is not due to using a larger model.

*Table 5.* Efficiency comparison in terms of runtime and memory usage. Each entry is reported as runtime (s)/memory (GB).

| Dataset | RecFormer | RPCIC | DMVG | PMIMC | Ours |
|---|---|---|---|---|---|
| Fashion | 470.8/2.5 | 899.5/1.9 | 820.7/1.6 | 928.7/1.8 | 853.3/1.6 |
| Scene-15 | 184.8/0.7 | 427.8/0.9 | 399.8/1.6 | 431.6/0.9 | 441.2/1.6 |

### 4.6. Different Missing-View Settings (Q6)

In the main experiments, incomplete views are generated with a shared missing rate across views, while the missing-view configurations vary among samples. To further assess the robustness of IMVC-NCD under different missing-view settings, we conduct additional experiments with two settings, PerV and SinV. Specifically, PerV simulates view-wise missingness by assigning each view an individual missing rate and randomly removing observations independently across views. SinV considers a more extreme missing pattern, where half of the samples remain complete across all views, while the remaining half retain only one identical observed view.

As shown in Table 6, IMVC-NCD achieves the best clustering performance in most cases under both different missing-view settings, further validating its effectiveness in handling diverse incomplete-view patterns.

*Table 6.* Clustering accuracy under different missing-view settings.

| Dataset | PerV (%) | | | SinV (%) | | |
|---|---|---|---|---|---|---|
| | PMIMC | DMVG | Ours | PMIMC | DMVG | Ours |
| HandWritten | 89.75 | 92.35 | **94.70** | 76.35 | 91.65 | **93.95** |
| Fashion | 70.43 | 80.34 | **83.92** | 72.73 | 78.20 | **79.19** |
| Aloi_deep | 88.90 | 90.45 | **91.82** | 87.56 | 89.62 | **91.42** |
| Scene-15 | 42.43 | 42.16 | **43.87** | 37.80 | 35.21 | 36.78 |

## 5. Conclusion

In this paper, we proposed the IMVC-NCD, a diffusion-driven latent completion framework that explicitly incorporates stable local neighborhood structures through the NCC module. By leveraging neighborhood-consistent and structurally reliable conditioning signals, IMVC-NCD enables stable, view-consistent latent recovery under missing-view patterns. Extensive experiments on multiple benchmark datasets demonstrate that our method consistently outperforms state-of-the-art IMVC approaches across diverse missing rates, highlighting the effectiveness of neighborhood-guided conditioning for diffusion-based latent reconstruction in incomplete multi-view learning. In future work, we will further explore more informative domain-aware structural priors and flexible joint optimization strategies, where the autoencoders can be updated together with the diffusion model while maintaining stable conditional guidance.

## Acknowledgements

This work was supported by National Natural Science Foundation of China (Nos. 62406218, 62306171) and the Special Fund for Science and Technology Innovation Teams of Shanxi Province (No. 202304051001001).

## Impact Statement

This paper presents work whose goal is to advance the field of Machine Learning. There are many potential societal consequences of our work, none which we feel must be specifically highlighted here.

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

# A. Appendix

## A.1. Datasets

Detailed descriptions of the datasets used for evaluation are provided below:

- **HandWritten:**[1] This image dataset, widely applicable across various domains, contains 2000 digital images of digits from 0 to 9, with 200 images of each digit. Each sample is represented by five distinct feature sets, namely 216-dimensional FAC, 76-dimensional FOU, 64-dimensional KAR, 240-dimensional Pix, and 47-dimensional ZER (LeCun et al., 1989).

- **Scene-15:** The dataset consists of 15 different scene categories, encompassing both indoor and outdoor environments, totaling 4485 images. Each image is represented by three extracted features, i.e., GIST, PHOG, and LBP, forming three distinct views (Fei-Fei & Perona, 2005).

- **Fashion:** This image dataset focuses on products, and following the methodology in Xu et al. (Xiao et al., 2017), we treat three different styles as three views of a single product. This dataset comprises 10 different objects, each with 1000 products, resulting in a total of 10,000 samples.

- **Aloi_deep:**[2] This is a multi-view dataset derived from the Aloi database, provided by RecFormer (Liu et al., 2023). This dataset includes 10,800 images selected from the Aloi database, encompassing 100 different objects, each representing a distinct cluster. The three-view features of this dataset are extracted by Liu et al. using three typical deep neural networks with pre-trained parameters, namely ResNet50 (He et al., 2016), VGG16 (Simonyan & Zisserman, 2014), and Inception-v3 (Szegedy et al., 2016).

## A.2. Compared Methods

To evaluate the effectiveness of the proposed IMVC-NCD, we compare it with ten state-of-the-art methods for incomplete multi-view clustering. We further categorize them into (i) *imputation-free* and (ii) *imputation-based* approaches, where the imputation-based methods are divided into *Non-DIMVC* (traditional imputation) and *DIMVC* (deep imputation). Moreover, DIMVC methods are grouped into *graph-structured*, *prototype-based*, and *generative-model* paradigms. The compared methods are listed as follows.

- **Imputation-free methods:**
  - **SIMVC-SA (Wen et al., 2023):** It performs IMVC by constructing view-specific anchor graphs to handle missing-views efficiently. By aligning the anchor-based graph structures across different views, SIMVC-SA achieves scalable and effective clustering under incomplete multi-view settings.

- **Imputation-based methods:**
  - **Non-DIMVC (traditional imputation):**
    * **BSV (Zhao et al., 2016):** It is a simple baseline method that fills missing-views with the average vector and performs $k$-means clustering independently on each view to obtain the final results.
    * **Concat (Zhao et al., 2016):** It adopts the same mean-value imputation strategy as BSV and simply concatenates all views to conduct single-view clustering.
  - **DIMVC (deep learning-based imputation):**
    * **Graph-structured DIMVC:**
      · **RecFormer (Liu et al., 2023):** It addresses IMVC by learning cross-view representations with a Transformer-based autoencoder framework. RecFormer reconstructs missing-views via recurrent graph-based information propagation, enabling effective clustering under incomplete multi-view settings.
    * **Prototype-based DIMVC:**
      · **CPSPAN (Jin et al., 2023):** It is a deep IMVC method that performs cross-view partial sample alignment to handle missing-views. By jointly aligning samples and prototypes across views, CPSPAN enhances cross-view consistency and improves clustering performance under incomplete data.

---

[1] https://archive.ics.uci.edu/ml/datasets/Multiple+Features
[2] https://github.com/justsmart/Recformer

- · **RPCIC (Yuan et al., 2024a):** It is a prototype-based IMVC method designed to improve robustness under missing-views. By learning and aligning cross-view prototypes, RPCIC mitigates the influence of incomplete data and achieves robust clustering performance.
- · **PMIMC (Yuan et al., 2025):** It addresses IMVC by explicitly matching cross-view prototypes and performing prototype-based missing-view imputation. Unlike RPCIC, which emphasizes robustness through prototype alignment, PMIMC focuses on prototype matching and reconstruction to recover missing-views for clustering.
- ∗ **Generative model-based DIMVC:**
  - · **DMVG (Wen et al., 2024):** It is a diffusion-based IMVC method that models the data generation process to recover missing-views. By generating missing-views conditioned on the available ones through a diffusion model, DMVG enables effective clustering under incomplete multi-view settings.
  - · **BURG (Jin et al., 2025):** It is a normalizing flow-based DIMVC method designed for missing-view recovery via cross-view distribution transfer. BURG further introduces a dual-consistency recovery guidance mechanism (neighbor-aware and prototypical consistency) to preserve intra-view structures and enhance cross-view clustering alignment under missing-view patterns.
  - · **A$^2$CLN (Peng et al., 2025):** It is a GAN-based IMVC method that leverages adversarial feature generation to perform missing-view compensation and improve representation robustness. It further incorporates adaptive contrastive learning to promote cross-view consistency and then performs clustering in the fused latent space.

### A.3. Implementation Details

Our model is implemented based on PyTorch 2.4.0 and trained on a desktop computer equipped with an NVIDIA GeForce RTX 4090 GPU and 64 GB RAM.

**Autoencoder:** They are designed as view-specific multi-layer perceptrons (MLPs). For each view $v$, the encoder transforms the input feature space $d_v$ into a latent representation via the following architecture: $d_v \rightarrow 512 \rightarrow 512 \rightarrow 2048 \rightarrow 128$. ReLU activations are applied after each hidden layer. We apply batch normalization (BN) to the latent output, followed by layer normalization (LN) to stabilize the latent distributions across different views. The decoder creates a symmetric structure to the encoder, mapping the latent representations back to the original feature space: $128 \rightarrow 2048 \rightarrow 512 \rightarrow 512 \rightarrow d_v$, using fully connected layers with ReLU activations. Missing-views are explicitly handled by masking the corresponding input features during the encoding stage. All autoencoders are pretrained for 200 epochs. We adopt full-batch training to stabilize optimization and ensure consistent reconstruction signals across views, particularly under missing patterns.

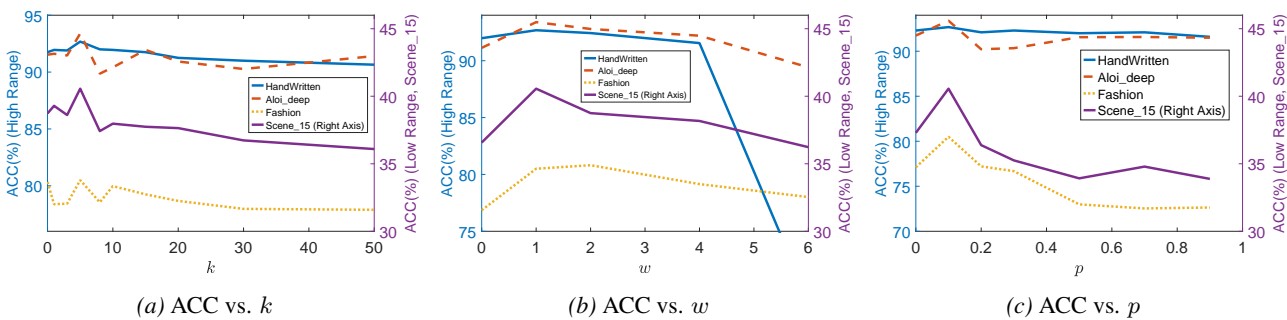

*(a)* ACC vs. $k$      *(b)* ACC vs. $w$      *(c)* ACC vs. $p$

*Figure 5.* ACC comparison under different hyperparameter settings under a missing rate of 50%: (a) varying $k$ in NCC; (b) varying guidance scale $w$; (c) varying dropout rate $p$.

**Conditional Diffusion-based Latent Completion**: We train view-specific DDPM independently within the latent space. The denoising network utilizes a 1-D U-Net architecture featuring residual convolutional blocks, hierarchical downsampling and upsampling paths, and skip connections. Specifically, temporal embeddings are injected at multiple resolution levels to encode diffusion time steps, while neighborhood-conditioned vectors are concatenated with latent features at the bottleneck layer to guide structure-aware generation. Each DDPM is trained for 500 epochs with a batch size of 512 and $T = 1000$ diffusion steps, employing a linear noise schedule ranging from $\beta_1 = 10^{-6}$ to $\beta_T = 2 \times 10^{-2}$. To enable classifier-free guidance, we randomly drop the conditional input with a probability of $p = 0.1$ during training, and subsequently apply a

*Table 7.* Comparison with generative models on HandWritten under different missing rates.

| Dataset | Method | ACC (30%) | ACC (50%) | ACC (70%) | NMI (30%) | NMI (50%) | NMI (70%) | Purity (30%) | Purity (50%) | Purity (70%) |
|---------|--------|-----------|-----------|-----------|-----------|-----------|-----------|--------------|--------------|--------------|
| HandWritten | MVAE | $91.05 \pm 0.29$ | $75.47 \pm 4.42$ | $44.52 \pm 1.80$ | $82.69 \pm 0.48$ | $67.40 \pm 3.08$ | $42.76 \pm 1.78$ | $91.05 \pm 0.29$ | $75.47 \pm 4.42$ | $44.92 \pm 1.69$ |
| | MMVAE | $91.05 \pm 0.05$ | $88.78 \pm 0.08$ | $84.15 \pm 0.05$ | $82.64 \pm 0.23$ | $79.35 \pm 0.05$ | $72.37 \pm 0.09$ | $91.05 \pm 0.05$ | $88.78 \pm 0.08$ | $84.15 \pm 0.05$ |
| | Ours | $\mathbf{94.56 \pm 0.12}$ | $\mathbf{92.68 \pm 0.05}$ | $\mathbf{87.55 \pm 0.15}$ | $\mathbf{88.46 \pm 0.11}$ | $\mathbf{85.34 \pm 0.21}$ | $\mathbf{77.13 \pm 0.21}$ | $\mathbf{94.56 \pm 0.12}$ | $\mathbf{92.68 \pm 0.05}$ | $\mathbf{87.55 \pm 0.15}$ |

*Table 8.* Comparison with generative models on Fashion, Aloi_deep, and Scene-15 under different missing rates.

| Dataset | Method | ACC (10%) | ACC (30%) | ACC (50%) | NMI (10%) | NMI (30%) | NMI (50%) | Purity (10%) | Purity (30%) | Purity (50%) |
|---------|--------|-----------|-----------|-----------|-----------|-----------|-----------|--------------|--------------|--------------|
| Fashion | MVAE | $80.93 \pm 2.34$ | $66.44 \pm 0.07$ | $38.82 \pm 0.70$ | $78.04 \pm 0.23$ | $63.67 \pm 0.33$ | $40.34 \pm 0.97$ | $83.14 \pm 0.13$ | $68.11 \pm 0.91$ | $40.17 \pm 0.22$ |
| | MMVAE | $80.25 \pm 0.03$ | $77.23 \pm 0.10$ | $64.10 \pm 0.08$ | $79.05 \pm 0.23$ | $74.56 \pm 0.38$ | $63.54 \pm 0.18$ | $80.25 \pm 0.03$ | $77.65 \pm 0.25$ | $63.34 \pm 0.04$ |
| | Ours | $\mathbf{87.56 \pm 0.29}$ | $\mathbf{85.52 \pm 0.15}$ | $\mathbf{80.50 \pm 0.29}$ | $\mathbf{82.66 \pm 0.36}$ | $\mathbf{79.25 \pm 0.07}$ | $\mathbf{72.88 \pm 0.04}$ | $\mathbf{87.56 \pm 0.29}$ | $\mathbf{85.52 \pm 0.15}$ | $\mathbf{80.50 \pm 0.29}$ |
| Aloi_deep | MVAE | $89.94 \pm 1.12$ | $84.96 \pm 1.61$ | $56.42 \pm 1.50$ | $97.04 \pm 0.04$ | $95.07 \pm 0.92$ | $82.55 \pm 0.41$ | $91.93 \pm 0.54$ | $87.49 \pm 1.53$ | $59.80 \pm 1.24$ |
| | MMVAE | $85.49 \pm 0.27$ | $87.99 \pm 1.32$ | $86.80 \pm 1.11$ | $95.21 \pm 0.03$ | $96.01 \pm 0.64$ | $95.64 \pm 0.42$ | $87.57 \pm 0.35$ | $90.16 \pm 1.35$ | $89.12 \pm 0.91$ |
| | Ours | $\mathbf{93.70 \pm 0.38}$ | $\mathbf{93.28 \pm 0.16}$ | $\mathbf{93.39 \pm 0.07}$ | $\mathbf{98.33 \pm 0.21}$ | $\mathbf{98.30 \pm 0.04}$ | $\mathbf{98.19 \pm 0.05}$ | $\mathbf{94.85 \pm 0.44}$ | $\mathbf{94.58 \pm 0.21}$ | $\mathbf{94.41 \pm 0.06}$ |
| Scene-15 | MVAE | $44.34 \pm 0.55$ | $35.57 \pm 0.36$ | $24.80 \pm 0.14$ | $41.21 \pm 0.87$ | $32.95 \pm 0.89$ | $24.71 \pm 0.41$ | $48.74 \pm 0.67$ | $39.14 \pm 0.77$ | $27.95 \pm 0.30$ |
| | MMVAE | $42.80 \pm 0.28$ | $34.57 \pm 0.66$ | $25.19 \pm 0.83$ | $43.50 \pm 0.09$ | $34.06 \pm 0.86$ | $25.87 \pm 0.88$ | $46.58 \pm 0.41$ | $38.03 \pm 1.21$ | $28.26 \pm 0.62$ |
| | Ours | $\mathbf{45.47 \pm 0.55}$ | $\mathbf{43.38 \pm 0.36}$ | $\mathbf{40.56 \pm 0.83}$ | $\mathbf{45.01 \pm 0.34}$ | $\mathbf{42.00 \pm 0.15}$ | $\mathbf{37.24 \pm 0.32}$ | $\mathbf{49.32 \pm 0.76}$ | $\mathbf{47.71 \pm 0.16}$ | $\mathbf{45.29 \pm 0.30}$ |

guidance scale of $\omega = 1.0$ during inference. For structural guidance, neighborhood-conditioned vectors are derived from the $k = 5$ nearest neighbors in the latent space (based on Euclidean distance). Finally, the recovered and observed latent representations from all views are concatenated and clustered using the $k$-means algorithm.

### A.4. Additional Hyperparameter Analysis under 50% Missing Rate

We further examine the sensitivity of IMVC-NCD to three key hyperparameters under a 50% missing rate: the neighborhood size $k$, the classifier-free guidance scale $w$, and the dropout rate $p$. As shown in Fig. 5, the proposed method achieves relatively good performance when $k$ is set in the range of 1 to 8, $w$ is around 1 to 2, and $p$ is chosen around 0.1. These results confirm that IMVC-NCD maintains strong stability under different hyperparameter settings.

### A.5. Additional Comparisons with Generative Models

Following the reviewers' suggestion, we further compare IMVC-NCD with two general multi-view generative models, MVAE and MMVAE, to better demonstrate the advantage of the proposed neighborhood-conditioned diffusion framework. As shown in Tables 7 and 8, IMVC-NCD consistently achieves better clustering performance than these generative baselines across different datasets and missing rates. Although IMVC-NCD, MVAE, and MMVAE all belong to generative modeling methods, their objectives are different. MVAE and MMVAE mainly focus on general multimodal generation and joint/cross-modal generation, rather than preserving clustering-oriented structures under incomplete multi-view scenarios. In contrast, IMVC-NCD performs generation in a task-oriented manner for IMVC. By using neighborhood-conditioned recovery, our method explicitly preserves local structural consistency and produces clustering-friendly latent representations. Therefore, IMVC-NCD aligns better with the clustering objective and achieves superior performance.

