# OpenReview forum: "Incomplete Multi-View Clustering via Neighborhood-Conditioned Diffusion"
_ICML.cc/2026/Conference — ICML 2026 regular_

### Official Review · Reviewer_cBsd · 2026-02-15

**Soundness:** 2
**Presentation:** 3
**Significance:** 2
**Originality:** 2
**Overall Recommendation:** 4
**Confidence:** 3

**Summary:**

This paper proposes IMVC-NCD, a diffusion-based framework for incomplete multi-view clustering. The method first learns view-specific latent representations via autoencoders, then constructs neighborhood-conditioned vectors by aggregating cross-view local structures, and finally employs conditional diffusion models to recover missing-view latent representations. The completed multi-view embeddings are fused for clustering. The main contribution is introducing an explicit neighborhood-conditioned diffusion mechanism to improve robustness under heterogeneous missing-view scenarios, supported by empirical gains across several benchmark datasets.

**Compliance With Llm Reviewing Policy:**

Affirmed.

**Final Justification:**

Thank you for the detailed and constructive rebuttal. I appreciate the authors’ effort in addressing the concerns, particularly the additional experiments (e.g., comparisons with MVAE/MMVAE and runtime analysis) and the clarification on the role and limitations of neighborhood-conditioned guidance. These responses are helpful and strengthen the paper.

While some concerns regarding novelty and underlying assumptions remain, I find the empirical evidence and overall framework sufficiently convincing.

I will update my score to **4 (weak accept)**.

**Key Questions For Authors:**

* The key premise that local neighborhoods remain stable under heterogeneous missingness lacks supporting analysis (math analysis or failure-mode comparison), making it unclear when NCC is reliable.

* The method performs purely view-specific representation learning before cross-view generation, which may underutilize cross-view correlations; a comparison to joint multi-view encoders (e.g., MVAE/MMVAE) or an ablation is needed.

* It is unclear what advantages IMVC-NCD has over generic multi-view generative models (MVAE/MMVAE, MV-GPLVM), which also provide representation learning and missing-view generation at potentially lower cost. Including these baselines would strengthen the empirical claims.

* Computational complexity is under-discussed: training a DDPM per view with (T = 1000) steps is expensive; runtime/memory/scaling comparisons are needed.

**Limitations:**

Yes; the authors include an impact statement and, given the primarily methodological nature of the work, it is reasonable that no significant negative societal impact is identified.

**Strengths And Weaknesses:**

## Strengths

* Clear and intuitive idea: explicit neighborhood-conditioned diffusion for IMVC.
* Strong empirical performance across multiple missing-rate settings.
* Well-structured pipeline and clear presentation.
* Ablations indicate each component contributes to performance.

## Weaknesses

* Limited methodological novelty; largely a combination of existing components.
* Core assumption (neighborhood stability under heterogeneous missingness) lacks theoretical or empirical justification.
* View-specific representation learning may underutilize cross-view correlations; no comparison with joint multi-view encoders (e.g., MVAE/MMVAE).
* Missing comparisons with generic multi-view generative baselines (MVAE, MV-GPLVM), making advantages unclear.
* Computational cost of per-view diffusion models is under-discussed (no runtime or scaling analysis).

please see details below

---

> ### Author Rebuttal · Authors · 2026-03-27
>
> Thank you for insightful feedback. For brevity, data are denoted as follows: Fashion(F),  Scene15(S).
>
> **W1:** Our method focuses on ensuring that recovered views retain the original clustering structure as much as possible, thereby improving the results of the clustering task. This is our core innovation compared to compared methods. **Reviewer hr8f and wVVE** believes that our method "provides an innovative". The propsed neighborhood conditioning allows us to recover views that are valuable for clustering,even in noisy environments,rather than simply combining existing components. We validate superiority of our method  in noisy environments by comparing with two powerful baselines.Tab.1 shows our method ranks first, demonstrating its outstanding robustness to noise. Tab.2 indicate that the samples recovered by our method are closer to cluster centers, validating improvement in recovery quality.
>
> Tab.1:Noise robustness
> ||DMVG(%)||||Ours (%)||||
> |-|-:|-:|-:|-:|-:|-:|-:|-:|
> |noise_std|5|10|30|50|5|10|30|50|
> |F|76.23|76.87|77.55|73.15|80.33|80.52| 77.74|73.72|
> |S|33.98|26.6|13.60|10.50|34.83|33.31|17.44|12.82|
>
> Tab.2:Distance between recovered view and class centers
> ||DMVG||Ours||
> |-|-|-|-|-|
> |missing rate|30%|50%|30%|50%|
> |F|14.98|14.57|14.18|13.59|
> |S|11.77|11.67|8.79|8.10|
>
> **W2&Q1:** Thank you for this important comment. We agree that reliability of NCC under heterogeneous missingness should be better justified. Our claim is not that local neighborhoods remain strictly unchanged under all missing patterns, but that they are often more stable than pointwise cross-view correspondences and therefore provide a more reliable structural prior for clustering-oriented recovery. NCC is designed from this perspective by aggregating neighborhood cues from available views and encoding missingness states into the diffusion condition.
>
> Theoretically, clustering depends more on local relational structure than exact feature reconstruction. Under manifold assumption, local neighborhoods characterize intrinsic data geometry, preserving them helps maintain the affinity structure that supports downstream clustering. We also acknowledge that NCC has an applicability boundary, **its benefit may degrade when the k nearest neighbors include views from different clusters due to missing data(local neighborhood is corrupted)**.
>
> **W3&Q2:** In autoencoder stage, we ignore cross-view relationships to prevent the impact of varying view qualities on neighborhood condition construction.For example, the H dataset contains five views of varying quality, with clustering results of 74%, 58%, 73%, 59%, and 75% for each view, respectively. In the subsequent neighborhood conditions, each view is recovered using neighboring views, fully considering cross-view relationships to ensure recovery quality. To validate it, we replace the autoencoder with multi-view encoders (MVAE, MMVAE) for ablation. The results show using autoencoders independently on each view is more effective for constructing neighborhood conditions and better suited for clustering tasks.
>
> |Method|10%|30%|50%|
> |-|-|-|-|
> |Ours(MVAE)|85.18±2.34|82.16±0.07|63.66±0.70|
> |Ours(MMVAE)|90.70±0.03|88.46±0.10|87.52±0.08|
> |Ours|**93.70±0.38**|**93.28±0.16**|**93.39±0.07**|
>
> **W4&Q3:** This suggestion is very helpful in positioning IMVC-NCD relative to broader multi-view generative modeling approaches. While IMVC-NCD, MVAE, and MMVAE are all generative models, their goals differ. MVAE and MMVAE focus on general multimodal generative learning and joint/cross-modal generation, rather than preserving clustering structure. In contrast, IMVC-NCD uses generation in a task-oriented manner for incomplete multi-view clustering, where neighborhood-conditioned recovery process is specifically designed to preserve local structural consistency and produce clustering-friendly representations. As shown in the table below, IMVC-NCD aligns better with the clustering objective and achieves superior performance.
>
> |Data|Method|10%|30%|50%|
> |-|-|-|-|-|
> |F|MVAE|80.93±2.34|66.44±0.07|38.82±0.70|
> ||MMVAE|80.25±0.03|77.23±0.10|64.10±0.08|
> ||Ours|**87.56±0.29**|**85.52±0.15**|**80.50±0.29**|
> |S|MVAE|44.34±0.55|35.57±0.36| 24.80±0.14|
> ||MMVAE|42.80±0.28|34.57±0.66|25.19±0.83|
> ||Ours|**45.47±0.55**|**43.38±0.36**|**40.56±0.83**|
>
> **W5&Q4:** Thank you for your valuable suggestion. (X/X) in Tab. is time(s)/memory(GB) usage. As shown, diffusion-based methods do require more time and memory; however, current best-performing methods (DMVG, Ours) are also diffusion-based. Among these, our method has comparable time and memory consumption to strong baseline (DMVG) but outperforms it, indicating that superior performance is not due to a larger model.
>
> ||MVAE|MMVAE|RecFormer|RPCIC|DMVG|PMIMC|Ours|
> |-|-|-|-|-|-|-|-|
> |F|178.0/0.4|161.9/0.3|470.8/2.5|899.5/1.9|820.7/1.6|928.7/1.8|853.3/1.6|
> |S|66.8/0.3|69.9/0.3|184.8/0.7|427.8/0.9|399.8/1.6|431.6/0.9|441.2/1.6|
>
> We will include a more detailed analysis in the revised manuscript.

---

> > ### Author Rebuttal · Reviewer_cBsd · 2026-04-02
> >
> > Thank you for the detailed and constructive rebuttal. I appreciate the authors’ effort in addressing the concerns, particularly the additional experiments (e.g., comparisons with MVAE/MMVAE and runtime analysis) and the clarification on the role and limitations of neighborhood-conditioned guidance. These responses are helpful and strengthen the paper.
> >
> > While some concerns regarding novelty and underlying assumptions remain, I find the empirical evidence and overall framework sufficiently convincing.
> >
> > I will update my score to **4 (weak accept)**.

---

> > > ### Author Response · Authors · 2026-04-03
> > >
> > > Dear cBsd,
> > >
> > > We are pleased that we were able to address your concerns, and we sincerely appreciate your raising the score to 4.
> > >
> > > Best regards,
> > >
> > > The authors.

---

### Official Review · Reviewer_SQzh · 2026-03-06

**Soundness:** 2
**Presentation:** 2
**Significance:** 2
**Originality:** 2
**Overall Recommendation:** 2
**Confidence:** 4

**Summary:**

This paper does some research on incomplete multi-view clustering and proposes IMVC-NCD, a framework that combines view-specific latent learning, neighborhood-conditioned construction, and conditional diffusion-based latent completion. The core idea is to use local neighborhood structure from available views as a conditioning signal for recovering missing-view latent representations, followed by clustering on the completed multi-view embeddings. Experiments on four benchmark datasets show improved clustering performance over several prior IMVC baselines under the tested missing-view settings.

**Compliance With Llm Reviewing Policy:**

Affirmed.

**Final Justification:**

The authors partially resolved my problem.

**Key Questions For Authors:**

Please clarify how the claimed robustness to heterogeneous missingness is supported beyond the random masking protocol used in the experiments.

Please also clarify the exact implementation details for the baselines, especially any modified baselines, and provide a cleaner ablation that isolates the effect of neighborhood conditioning from the effect of diffusion itself.

Finally, a comparison of training and inference cost against the strongest baselines would be helpful.

**Limitations:**

The manuscript should discuss its limitations more explicitly, especially the realism of the missingness setting, the computational overhead of diffusion-based completion, and the limited benchmark diversity. The current impact statement is also very generic and does not meaningfully discuss possible limitations or broader implications.

**Strengths And Weaknesses:**

Strengths:
The paper solves a relevant IMVC problem and presents a coherent pipeline. The method in this paper is easy to follow at a high level, and the use of neighborhood-conditioned signals for latent completion is a reasonable design choice. The results are generally strong on the reported benchmarks, and this work also includes ablation and hyperparameter analysis.

Weaknesses:
The method appears to combine familiar components—autoencoders, kNN-style neighborhood aggregation, conditional diffusion, and guidance—into a new IMVC pipeline, but this manuscript does not fully establish a strong technical novelty beyond this combination. In addition, the missingness setup of the experiments seems to be based on standard random masking, which does not fully support broader claims about robustness to heterogeneous missingness. The scope of the evaluation is also limited, and the manuscript does not report runtime or memory cost despite using diffusion models.

---

> ### Author Rebuttal · Authors · 2026-03-27
>
> We thank you for the insightful comment, and if our clarification helps improve the understanding of our work, we would greatly appreciate your positive consideration. For brevity, datasets are denoted as follows: HandWritten(H), Fashion(F), Aloi_deep(A), Scene-15(S).
>
> **W1:** We appreciate your comments, but we respectfully disagree with the view that our method is simply a combination of familiar components. In IMVC, the quality of recovered views is crucial for downstream clustering performance. The main innovation of our method lies in the use of neighborhood conditions to ensure that the recovered views closely align with the original clustering structure, setting it apart from baseline approaches. Reviewers **hr8f** and **wVVE** clearly consider our method innovative, with hr8f describing it as "an innovative solution."
>
> Additionally, we further validate its innovation by comparing it with the strong baseline (DMVG) from two aspects: noise robustness and distance between the recovered views and class centers.
> As shown in Tab. 1, our method still ranks first, demonstrating strong robustness to noisy data. Tab. 2 further shows that the recovered samples are closer to the clustering centers, validating the improved recovery quality.
>
> Tab.1:Noise robustness
> ||DMVG(%)||||Ours (%)||||
> |-|-:|-:|-:|-:|-:|-:|-:|-:|
> |noise_std|5|10|30|50|5|10|30|50|
> |H|91.00|90.15|70.75|35.25|91.20|89.90|76.80|48.42|
> |F|76.23|76.87|77.55|73.15|80.33|80.52| 77.74|73.72|
> |A|88.57|88.54|85.11|74.05|89.52|92.85| 90.82|91.13|
> |S|33.98|26.6|13.60|10.50|34.83|33.31|17.44|12.82|
>
> Tab.2:Distance between the recovered views and class centers
> ||DMVG||Ours||
> |-|-|-|-|-|
> |missing rate|30%|50%|30%|50%|
> |H|18.74|17.88|18.39|17.38|
> |F|14.98|14.57|14.18|13.59|
> |A|9.58|9.43|7.71|8.17|
> |S|11.77|11.67|8.79|8.10|
>
> **W2&Q1:** In the process of constructing missing views, the missing rate is the same across different views, but varies across different samples, leading to heterogeneous missingness. Your suggestion is insightful and based on them, we further conducted experiments under other heterogeneous missingness scenarios (**PerV**  and **SinV**).  These results are reported in the below table.  Specifically, **PerV** simulates view-wise missingness by assigning each view an individual missing rate and randomly removing observations independently across views, while **SinV** has devised a more extreme missing data configuration, where half of the samples maintain complete visibility across all views, while the remaining half retain only one identical view.
>
> As shown in Tab. 3, our method performs excellently under different types of heterogeneous missingness, achieving the best clustering performance in most cases, which validates its effectiveness in handling such issues.
>
> Tab.3:Clustering results in other heterogeneous missingness scenarios
>
> ||perV(%)|||SinV(%)|||
> |-|-|-|-|-|-|-|
> |Datasets|PMIMC|DMVG|Ours|PMIMC|DMVG|Ours|
> |H|89.75|92.35|**94.70**|76.35|91.65|**93.95**|
> |F|70.43|80.34|**83.92**|72.73|78.20|**79.19**|
> |A|88.90|90.45|**91.82**|87.56|89.62|**91.42**|
> |S|42.43|42.16|**43.87**|**37.80**|35.21|36.78|
>
> **W3&Q2:** We apologize for the lack of clarity. We only modified the baseline methods in terms of input and output to align with the dataset's requirements. For the strong baseline (DMVG), we made adjustments to ensure that its components are consistent with ours for a fair comparison. The only difference lies in the diffusion conditioning. The diffusion condition of DMVG is to concatenate all views directly. The table below is provided to disentangle the effect of neighborhood conditioning (NC) from that of the diffusion model itself.
>
> It can be seen from the table that the use of neighborhood conditioning in diffusion significantly improves the model’s performance, indicating that neighborhood conditioning has a substantial impact on the final results.
>
> Tab.4:Effects of neighborhood conditioning and those of the diffusion model itself
> ||H(%)||F(%)||
> |-|-|-|-|-|
> |missing rate|30|50|30|50|
> |w/o NC|79.8|52.1|81.8|49.7|
> |NC|94.6|92.7|87.6|85.5|
>
> **W4&Q3:** We add the training and inference times with strongest baseline(DMVG). Our method takes slightly longer to train, while inference time is similar to DMVG. Although the training time is slightly longer, the significant improvement in clustering performance makes this extra cost accept.
>
> Tab.5:Training time and inference time
> ||DMVG(s)||Ours(s)||
> |-|-|-|-|-|
> |datasets|train|inference|train|inference|
> |H| 229.3|86.7|254.9|86.0|
> |F| 589.3|225.6|622.3|225.1|
> |A| 638.4|261.7|670.1|260.0|
> |S| 294.9|101.7|336.4|101.2|
>
> Your suggestions have greatly improved the quality of our manuscript, and we would be very happy to incorporate them into the final version.

---

> > ### Author Rebuttal · Reviewer_SQzh · 2026-04-02
> >
> > Thank you for the rebuttal. I appreciate the added experiments and runtime comparison. They improve the empirical support for the paper.
> >
> > However, my concerns are only partially resolved. The rebuttal strengthens the case for effectiveness, but it does not fully resolve my originality concern. Also, the fairness of the comparison with DMVG remains somewhat unclear because the baseline is evaluated in a modified/aligned form.
> >
> > Follow-up questions for the authors:
> > Please clarify how the modified DMVG differs from the original method, and what you see as the main technical novelty beyond combining existing components.

---

> > > ### Author Response · Authors · 2026-04-04
> > >
> > > We sincerely thank the reviewer for this valuable comment. As pointed out, we did make certain adjustments to the baseline method DMVG in our experiments. In the early stage of our method design, inspired by BURG and A²CLN, we first adopted a multilayer perceptron (MLP)-based autoencoder to learn representations from multi-view data. Later, during our reproduction of DMVG, we found that its original implementation employed a U-Net-like convolutional autoencoder for computational efficiency. However, unlike image generation tasks, representation learning and diffusion-based recovery in IMVC are typically conducted in a low-dimensional latent space rather than in the original observation space with strong spatial locality. Since such latent representations usually do not exhibit clear local neighborhoods or translation invariance, an MLP-based architecture is generally more suitable than a convolutional one for this task. On the one hand, MLPs are more conducive to global interactions across feature dimensions; on the other hand, they are more convenient for integrating conditional information from heterogeneous views. Based on these considerations, to minimize the additional influence caused by differences in encoder architecture and make a comparison as fair as possible, we equipped DMVG with the same MLP-based autoencoder as used in our method. In other words, our adjustment to DMVG was primarily intended to improve its suitability for the current task setting while reducing the extra impact introduced by encoder architectural differences.
> > >
> > > To further address the reviewer’s concern,  based on your professional suggestion, we additionally report  the results of the **original DMVG (ODMVG) with any modification**, the original DMVG with our proposed neighborhood conditioning (ODMVG+NC), DMVG equipped with our autoencoder (DMVG), and our full method. The experimental results  in Tables 1 and 2 show that the original DMVG setting performs noticeably worse than the version using our autoencoder. At the same time, the constructed neighborhood-conditioned guidance further improves clustering performance.
> > >
> > > Overall, we believe that the comparison between our method and DMVG is fair and reasonable.
> > >
> > >
> > > Table 1: Performance comparison with DMVG variants on the HandWritten dataset.
> > > |Dataset|method|ACC(30%)|ACC(50%)|ACC(70%)|
> > > |-|-|-|-|-|
> > > |HandWritten|ODMVG|92.98±0.27|90.78±0.57|84.90±0.65|
> > > ||ODMVG+NC|93.25±0.02|91.45±0.25|85.15±0.30|
> > > ||DMVG|93.55±0.02|91.66±0.69|86.77±0.60|
> > > ||Ours|**94.56±0.12**|**92.68±0.05**|**87.55±0.15**|
> > >
> > > Table 2: Performance comparison with DMVG variants on the Scene-15 dataset.
> > > |Dataset|method|ACC(10%)|ACC(30%)|ACC(50%)|
> > > |-|-|-|-|-|
> > > |Scene-15|ODMVG|42.24±0.93|39.96±0.13|36.25±1.01|
> > > ||ODMVG+NC|42.81±0.88|41.55±0.43|37.37±1.13|
> > > ||DMVG|43.55±0.04|41.26±0.99|37.41±0.27|
> > > ||Ours|**45.47±0.55**|**43.38±0.36**|**40.56±0.83**|
> > >
> > > The core innovation of our method lies in the construction of the neighborhood conditions for the diffusion model. This carefully designed condition guides the diffusion model to recover missing views while maximally preserving the intrinsic clustering structure, thereby directly boosting the downstream clustering performance. This represents the fundamental distinction between our approach and DMVG. Rather than simply cascading existing components, our proposed domain condition empowers the model to recover views that are highly valuable for clustering, even under noisy environments.
> > > Table 3 shows our method ranks first, demonstrating its outstanding robustness to noise. Table 4 indicate that the samples recovered by our method are closer to cluster centers, validating improvement in recovery quality.
> > >
> > > Table 3: Noise robustness
> > > ||DMVG(%)||||Ours (%)||||
> > > |-|-|-|-|-|-|-|-|-|
> > > |noise_std|5|10|30|50|5|10|30|50|
> > > |Fashion|76.23|76.87|77.55|73.15|80.33|80.52|77.74|73.72|
> > > |Scene-15|33.98|26.60|13.60|10.50|34.83|33.31|17.44|12.82|
> > >
> > > Table 4: Distance between recovered view and class centers
> > > ||DMVG||Ours||
> > > |-|-|-|-|-|
> > > |missing rate|30%|50%|30%|50%|
> > > |Fashion|14.98|14.57|14.18|13.59|
> > > |Scene-15|11.77|11.67|8.79|8.10|
> > >
> > > We thank you for the insightful comment again, and if our clarification helps improve the understanding of our work, we would greatly appreciate your positive consideratio

---

### Official Review · Reviewer_wVVE · 2026-03-08

**Soundness:** 3
**Presentation:** 4
**Significance:** 4
**Originality:** 3
**Overall Recommendation:** 5
**Confidence:** 4

**Summary:**

The paper proposed an innovative neighborhood-conditioned diffusion for incomplete multi-view clustering (IMVC-NCD). As a generative model, the quality of the diffusion conditions in diffusion models has a significant impact on the effectiveness of completing missing views. IMVC-NCD is the first method to introduce neighborhood conditions as diffusion constraints within a diffusion-based framework. By leveraging the local structure of neighborhood relationships in the data, it enhances the stability and accuracy of the diffusion process, effectively mitigating the issue of incomplete information caused by missing views in multi-view clustering. This approach not only improves clustering performance but also provides new insights for the field of multi-view learning.

**Compliance With Llm Reviewing Policy:**

Affirmed.

**Final Justification:**

The authors have provided additional experiments and a clear explanation, which have addressed all of my concerns. Therefore, I am raising my score to 5.

**Key Questions For Authors:**

The authors have already analyzed the sensitivity of hyperparameters, but it is recommended to conduct more experiments to investigate the impact of hyperparameters under different missing view ratios. This will help to more comprehensively evaluate the model's performance and stability.
If the authors address my concerns, I am happy to increase my score.

**Limitations:**

yes

**Strengths And Weaknesses:**

Strength:
(1) The approach is innovative and effectively alleviates the instability in view recovery caused by weak structural constraints. Its premise is clear and forward-thinking.
(2) The experiments are comprehensive and validate the generalization ability and superiority of the IMVC-NCD from multiple perspectives.
(3) The paper is logically organized, and I really enjoy reading.

Weaknesses:
(1) The comparison method DMVG is also a diffusion-based generative method. So, the authors need to explain the implementation of DMVG and analyze the reasons behind the performance advantages of IMVC-NCD.
(2) It is recommended to provide more experimental details, such as whether end-to-end training was used. This would improve code reusability and the scalability of the experiments.

---

> ### Author Rebuttal · Authors · 2026-03-30
>
> We thank you for the insightful comment, and if our clarification helps improve the understanding of our work, we would greatly appreciate your positive consideration. For brevity, datasets are denoted as follows: HandWritten(H), Fashion(F), Aloi_deep(A), Scene-15(S).
>
> **Q1:**  We sincerely thank the reviewer for this helpful suggestion. Based on your suggestion, we provide additional hyperparameter sensitivity analyses at a missing rate of 0.5, and the experimental results are summarized in the tables below.
>
> Specifically, we examine the sensitivity of IMVC-NCD to three important hyperparameters: the neighborhood size $k$ in the NCC module, the classifier-free guidance scale $w$, and the dropout rate $p$. The results show that the proposed method achieves relatively good performance when $k$ is set in the range of 1 to 8, $w$ is set around 1 to 2, and $p$ is chosen around 0.1. These results confirm that IMVC-NCD maintains strong stability under different hyperparameter settings.
>
> Table1: Sensitivity analysis of neighborhood size $k$
>
> |$k$|0|1|3|5|8|10|15|20|30|50|
> |-|-|-|-|-|-|-|-|-|-|-|
> |H|91.75|91.95|91.90|92.68|92.00|91.95|91.75|91.25|91.00|90.65|
> |F|80.33|78.38|78.43|80.50|78.55|79.97|79.25|78.67|77.97|77.89|
> |A|91.53|91.61|91.46|93.39|89.86|90.42|91.95|90.93|90.27|91.41|
> |S|38.73|39.29|38.61|40.56|37.42|37.97|37.74|37.64|36.73|36.09|
>
> Table2: Sensitivity analysis of guidance scale $w$
>
> |$w$|0|1|2|4|6|
> |-|-|-|-|-|-|
> |H|91.98|92.68|92.43|91.55|68.95|
> |F|76.85|80.50|80.81|79.15|78.01|
> |A|91.13|93.39|92.79|92.20|89.42|
> |S|36.57|40.56|38.75|38.17|36.23|
>
> Table3: Sensitivity analysis of dropout rate $p$
>
> |$p$|0|0.1|0.2|0.3|0.5|0.7|0.9|
> |-|-|-|-|-|-|-|-|
> |H|92.30|92.68|92.10|92.30|92.00|92.10|91.60|
> |F|77.11|80.50|77.21|76.68|72.99|72.54|72.64|
> |A|91.74|93.39|90.22|90.34|91.56|91.58|91.50|
> |S|37.29|40.56|36.38|35.25|33.92|34.79|33.88|
>
> **W1:** Thank you for this insightful comment. We agree that, since DMVG is also a diffusion-based generative method, its implementation and the difference from our method should be clarified more explicitly.
>
> DMVG is essentially a conditional diffusion model for missing-view generation. Specifically, for each target view, it trains a separate U-Net-based diffusion model that takes the noisy target view and the remaining available views as conditions, and then predicts the injected noise for iterative denoising. It is mainly designed to recover the original missing views and then serve as a plug-in module for downstream multi-view clustering. It also considers full-, partial-, and no-condition cases, and further adopts classifier-free guidance during training and inference.
>
> Both DMVG and IMVC-NCD are developed for missing-view generation; however, the superior performance of IMVC-NCD mainly stems from its neighborhood-conditioned generation mechanism. By incorporating neighborhood-conditioned information into the recovery process, IMVC-NCD can exploit local structural relationships among samples, so that the generated views not only fill in the missing information but also better preserve the underlying semantic and cluster structure of the data. Consequently, the recovered views are more suitable for downstream clustering, which explains the performance gain over DMVG.
>
> **W2:** Thank you for this valuable suggestion. We would like to clarify that our method does not use end-to-end training. Instead, we first pretrain the view-specific autoencoders, then freeze their parameters, construct the neighborhood-conditioned information based on the fixed latent embeddings, and finally train the DDPM module. We adopt this two-stage training strategy for the following reasons:
>
> (1) Freezing the autoencoders provides a stable latent space for neighborhood construction and diffusion-based recovery. If the encoder parameters keep changing during DDPM training, the latent representations and their neighborhood relations would continuously drift, making the conditioning information unstable and hindering the diffusion model from learning reliable recovery patterns;
>
> (2) This design decouples representation learning from missing-view generation, which makes the training process more stable and easier to reproduce;
>
> (3) Training the diffusion model on fixed latent embeddings also reduces the difficulty of joint optimization and improves the overall scalability of the framework, since the DDPM can focus specifically on missing-view recovery in a stable latent space.

---

> > ### Author Rebuttal · Reviewer_wVVE · 2026-04-01
> >
> > The authors have provided additional experiments and a clear explanation, which have addressed all of my concerns. Therefore, I am raising my score to 5.

---

> > > ### Author Response · Authors · 2026-04-04
> > >
> > > Dear wVVE,
> > >
> > > We are pleased to have addressed your concerns and sincerely appreciate your positive feedback.
> > >
> > > Best regards,
> > >
> > > The authors.

---

### Official Review · Reviewer_hr8f · 2026-03-10

**Soundness:** 3
**Presentation:** 3
**Significance:** 4
**Originality:** 3
**Overall Recommendation:** 5
**Confidence:** 4

**Summary:**

The authors discovered that, in the case of heterogeneous missing data, local neighborhood relationships remain relatively stable. Based on this observation, they propose the neighborhood-conditioned diffusion (IMVC-NCD) to address the issue of incomplete multi-view clustering. This algorithm effectively integrates view-specific latent learning with structure-aware generative recovery, generating discriminative latent features with stronger clustering separability, which directly enhances the performance of subsequent clustering tasks. This method makes a valuable contribution to the field of incomplete multi-view clustering and provides a new framework supported by reliable experimental validation, with significant research implications.

**Compliance With Llm Reviewing Policy:**

Affirmed.

**Final Justification:**

The author's response addressed my main concerns, so I have decided to keep my rating.

**Key Questions For Authors:**

Please refer to the Weaknesses.

**Limitations:**

Yes

**Strengths And Weaknesses:**

Strength:
1. The paper is well-structured, with a clear logical flow and distinct sections. The methodology is described in detail and is easy to understand.
2. The experiments offer an extensive comparison, with beautifully designed charts that effectively support the arguments and enhance the visualization of the results.
3. The proposed neighborhood-conditioned diffusion framework offers an innovative approach to solving the incomplete multi-view clustering problem, demonstrating exceptional robustness, especially when dealing with severely missing views. This framework is expected to inspire researchers to explore more effective diffusion condition constraints, with significant potential impact and practical value, further advancing the development of this field.

Weaknesses:
1. Although the paper is generally well-written, the methodology section (Section 3.3) may not be sufficiently clear for readers unfamiliar with diffusion models. It is better to provide additional clarification.
2. Although the authors' experimental section is comprehensive, it could be further enhanced by adding more generative models for comparison, thereby highlighting the advantages of IMVC-NCD.

---

> ### Author Rebuttal · Authors · 2026-03-30
>
> We thank you for the insightful comment, and if our clarification helps improve the understanding of our work, we would greatly appreciate your positive consideration.
>
> **W1:** We thank the reviewer for this valuable comment. To improve the clarity of Section 3.3 for readers who may be less familiar with diffusion models, we will insert the following paragraph between the first and second sentences of Section 3.3. In addition, if the paper is accepted, we will add a clearer description of DDPM in the appendix.
>
> Originally, DDPM was introduced as a generative framework that learns to recover structured data, such as images, from progressively corrupted noise through a step-by-step denoising process. In our method, rather than generating images in the original data space, we transfer this mechanism to the latent space for IMVC, where the goal is to recover the missing latent representation of a target view from noise under the guidance of neighborhood-conditioned information.
>
> **W2:** Following your suggestion, we provide more compared experements to highlight the advantages of IMVC-NCD by comparing with more generative models MVAE and MMVAE. As shown in Table 1 and Table 2, IMVC-NCD consistently achieves better overall clustering performance than the general multi-view generative models MVAE and MMVAE.
>
> Table1: Comparison of clustering performance on HandWritten under different missing rates.
>
> |Dataset|Method|ACC(30%)|ACC(50%)|ACC(70%)|NMI(30%)|NMI(50%)|NMI(70%)|Purity(30%)|Purity(50%)|Purity(70%)|
> |--|--|--|--|--|--|--|--|--|--|--|
> |HandWritten|MVAE|91.05±0.29|75.47±4.42|44.52±1.80|82.69±0.48|67.40±3.08|42.76±1.78|91.05±0.29|75.47±4.42|44.92±1.69|
> ||MMVAE|91.05±0.05|88.78±0.08|84.15±0.05 |82.64±0.23|79.35±0.05|72.37±0.09|91.05±0.05|88.78±0.08|84.15±0.05|
> || Ours | **94.56±0.12**|**92.68±0.05**|**87.55±0.15**|**88.46±0.11**|**85.34±0.21**|**77.13±0.21**|**94.56±0.12**|**92.68±0.05**|**87.55±0.15**|
>
> Table2: Comparison of clustering performance on Fashion, Aloi_deep, and Scene-15 under different missing rates.
>
> |Dataset |Method|ACC(10%)|ACC(30%)|ACC(50%)|NMI(10%)|NMI(30%)|NMI(50%)|Purity(10%)|Purity(30%)|Purity(50%)|
> |--|--|--|--|--|--|--|--|--|--|--|
> |Fashion|MVAE|80.93±2.34|66.44±0.07|38.82±0.70|78.04±0.23|63.67±0.33|40.34±0.97|83.14±0.13|68.11±0.91|40.17±0.22|
> ||MMVAE|80.25±0.03|77.23±0.10|64.10±0.08|79.05±0.23|74.56±0.38|63.54±0.18|80.25±0.03|77.65±0.25|63.34±0.04|
> ||Ours|**87.56±0.29**|**85.52±0.15**|**80.50±0.29**|**82.66±0.36**|**79.25±0.07**|**72.88±0.04**|**87.56±0.29**|**85.52±0.15**|**80.50±0.29**|
> |Aloi_deep|MVAE|89.94±1.12|84.96±1.61|56.42±1.50|97.04±0.04|95.07±0.92|82.55±0.41|91.93±0.54|87.49±1.53|59.80±1.24|
> ||MMVAE|85.49±0.27|87.99±1.32|86.80±1.11|95.21±0.03|96.01±0.64|95.64±0.42|87.57±0.35|90.16±1.35|89.12±0.91|
> ||Ours|**93.70±0.38**|**93.28±0.16**|**93.39±0.07**|**98.33±0.21**|**98.30±0.04**|**98.19±0.05**|**94.85±0.44**|**94.58±0.1**|**94.41±0.06**|
> |Scene-15|MVAE|44.34±0.55|35.57±0.36| 24.80±0.14|41.21±0.87|32.95±0.89|24.71±0.41|48.74±0.67|39.14±0.77|27.95±0.30|
> ||MMVAE|42.80±0.28|34.57±0.66|25.19±0.83|43.50±0.09|34.06±0.86|25.87±0.88|46.58±0.41|38.03±1.21|28.26±0.62|
> ||Ours|**45.47±0.55**|**43.38±0.36**|**40.56±0.83**|**45.01±0.34**|**42.00±0.15**|**37.24±0.32**|**49.32±0.76**|**47.71±0.16**|**45.29±0.30**|

---

> > ### Author Rebuttal · Reviewer_hr8f · 2026-04-03
> >
> > Thank you for the detailed rebuttal by the author. It has completely resolved my doubts. I will maintain my rating.

---

> > > ### Author Response · Authors · 2026-04-03
> > >
> > > Dear hr8f,
> > >
> > > We are pleased to have addressed your concerns and sincerely appreciate your positive feedback.
> > >
> > > Best regards,
> > >
> > > The authors.

---

### Decision · Program_Chairs · 2026-04-30

**Decision:**

Accept (regular)

**Comment:**

This paper introduces neighborhood-conditioned diffusion for incomplete multi-view clustering (IMVC-NCD), which increases the clustering performance under the incomplete multi-view scenario through view-specific latent learning with structure-aware generative recovery. The reviewers generally agreed on the motivation and presentation. The reviewers' concerns about baseline comparisons and computational cost have been addressed in the rebuttal by providing related experiments. However, there remains some doubt on the technical novelty, which concerns the framework that combines existing components (reviewer SQzh). Given that the experiment results are solid, but there are still some concerns about novelty, I recommend this paper as Weak Accept.